# Scaling Laws for Online Advertisement Retrieval

## Abstract

The scaling law is a notable property of neural network models and has significantly propelled the development of large language models. Scaling laws hold great promise in guiding model design and resource allocation. Recent research increasingly shows that scaling laws are not limited to NLP tasks or Transformer architectures; they also apply to domains such as recommendation. However, there is still a lack of literature on scaling law research in online advertisement retrieval systems. This may be because 1) identifying the scaling law for resource cost and online revenue is often expensive in both time and training resources for industrial applications, and 2) varying settings for different systems prevent the scaling law from being applied across various scenarios. To address these issues, we propose a lightweight paradigm to identify online scaling laws of retrieval models, incorporating a novel offline metric $R/R^*$ and an offline simulation algorithm. We prove that under some assumptions, the correlation between $R/R^*$ and online revenue asymptotically approaches 1 and empirically validates its effectiveness. The simulation algorithm can estimate the machine cost offline. Based on the lightweight paradigm, we can identify online scaling laws for retrieval models almost exclusively through offline experiments, and quickly estimate machine costs and revenues for given model configurations. We further validate the existence of scaling laws across mainstream model architectures—including Transformer, MLP, and DSSM—in our real-world advertising system. With the identified scaling laws, we demonstrate practical applications for ROI-constrained model designing and multi-scenario resource allocation in the online advertising system. To the best of our knowledge, this is the first work to study the identification and application of online scaling laws for online advertisement retrieval, showing great potential for scaling laws in advertising system optimization.

## 1 Introduction

The neural scaling laws, describing how neural network performance changes with key factors (e.g. model size, dataset size, computational cost), have been discovered in various research areas (Kaplan et al., 2020; Hoffmann et al., 2022; Shin et al., 2023; Isik et al., 2024; Zhang et al., 2024; Fang et al., 2024). Early research (Hestness et al., 2017) shows that the neural network performance is predictable when scaling training data size in various tasks such as neural machine translation and language modeling. Kaplan et al.(2020) further empirically verify the scaling law of the Transformer architecture in language modeling, regarding the key factors (model size, data size, training cost) and training performance (PPL). Inspired by the scaling law, researchers extend the size of pre-trained language models and further empirically verify the scaling law by training GPT-3 (Brown, 2020). This wave of enthusiasm has led to the creation of GPT-3.5 and GPT-4 (Achiam et al., 2023), ushering in a new era of NLP research and applications.

Based on the scaling laws, the optimal key factors of the model can be determined under given constraints, thus guiding us in model design and resource allocation. Recommendation and advertising systems are mature commercial applications that prioritize ROI, making it highly valuable to explore whether there exist scaling laws for recommendation and advertising models. Due to the lack of a thriving community and open data ecosystem similar to NLP, research on model scaling is relatively scarce in the recommendation and advertisement areas. Early studies primarily gave some qualitative or offline quantification conclusions about model scaling (Shin et al., 2023; Zhang

et al., 2024). Fang et al.(2024) attempts to give a quantitative scaling law of model performance and the amount of training data and model parameters based on public information retrieval benchmarks, and first gives an offline application practice in the recommendation area, which is to solve the optimal amount of data and model parameters under a given total training resource.

Crucially, **no work has addressed the online identification of scaling laws between business revenue and machine costs in real-world advertising systems**. We attribute this to **two primary challenges**: 1) The scaling law must describe the online revenue-cost relationship (not offline metrics), requiring costly online experiments; 2) System-specific configurations prevent universal applicability, making scaling law adoption prohibitively expensive. As a result, to use scaling laws to guide system optimization in a specific scenario, one must first incur substantial costs to obtain the necessary parameters. This makes it impractical for commercial systems to identify and apply scaling laws in many real-world settings.

We aim to identify scaling laws in online advertising systems with low experimental costs and explore their applications in guiding system optimization. An advertising system typically consists of several subsystems (e.g., advertisement retrieval, bidding, and ranking), each with distinct technical architectures. This heterogeneity complicates establishing a unified paradigm for lightweight scaling law identification. In this work, we focus on the online advertisement retrieval subsystem—a critical component that selects the top-k ads for downstream stages without directly determining billing. This subsystem is described in detail in Section 2.

To address these challenges, we propose a **lightweight paradigm** for identifying online scaling laws in industrial advertisement retrieval. First, we introduce a novel offline metric, $R/R^*$, which is evaluated on training data that is an i.i.d. sample from the full-stage online system. It directly measures, offline, the ratio between the total value (eCPM) of the top-m ads selected by the model and the total value of the ground-truth top-m ads. We prove that, under some assumptions, the correlation between $R/R^*$ and online revenue asymptotically approaches 1. This theoretical guarantee, combined with historical A/B test data from daily model iterations, enables us to calibrate the $R/R^*$–revenue relationship without requiring additional online experiments. Then, we design a simulation algorithm to estimate machine cost from model configurations. By treating model size as the decision variable and using FLOPs and $R/R^*$ as intermediate proxies, our paradigm enables end-to-end prediction of both machine cost and online revenue, facilitating efficient offline model selection and system optimization.

We validate the effectiveness of $R/R^*$ through extensive online A/B tests. Results demonstrate that $R/R^*$ serves as a more accurate offline surrogate for online revenue than traditional metrics. Then, we conduct extensive offline experiments across mainstream retrieval architectures—Transformer (Vaswani et al., 2017), MLP (mlp, 1958), and DSSM (Huang et al., 2013)—in both the pre-ranking and matching stages. Using FLOPs as the independent variable and $R/R^*$ as the dependent variable, we observe consistent broken neural scaling laws (Caballero et al., 2023) across all architectures and stages. Building on this, we apply the identified scaling laws to two critical use cases: **ROI-constrained model design** and **multi-scenario resource allocation**. These deployments yield a substantial **combined 5.10% improvement in online revenue**, demonstrating the practical utility of scaling laws in real-world system optimization.

Our contributions are threefold: 1) We propose $R/R^*$, a novel offline metric that is both **theoretically justified** (asymptotically correlated with online revenue under some assumptions) and **empirically validated** as a superior surrogate for online revenue. 2) We present a lightweight paradigm for identifying scaling laws between machine cost and online revenue in industrial retrieval systems, enabling **offline discovery** of broken neural scaling laws across diverse architectures (e.g., Transformer, MLP, DSSM). 3) We demonstrate real-world applications in ROI-constrained model selection and multi-scenario resource allocation, achieving a substantial **5.10% gain in online revenue**. Our framework enables rapid offline ROI estimation for hundreds of model configurations within days—accelerating model iteration and system optimization at scale.

## 2 FORMULATION OF ONLINE ADVERTISEMENT RETRIEVAL

In this section, we clarify the research subject and experimental background of this paper. **We give a formulation of online advertisement retrieval, including the definition of the retrieval stage,**

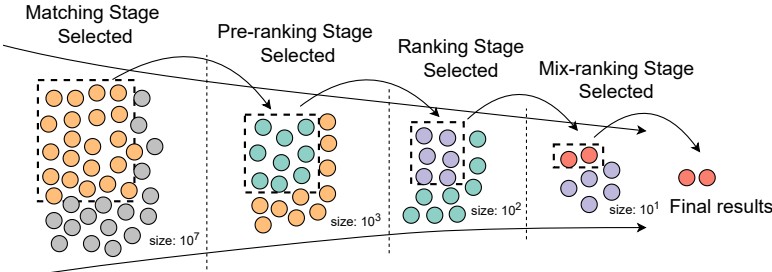

Figure 1: The cascade ranking architecture of our advertising system, includes four stages: Matching, Pre-ranking, Ranking, and Mix-ranking. Typically, we regard the "Matching" and "Pre-ranking" stages as retrieval stages, which do not decide the billing of ads.

**mainstream industrial practice of architecture and learning tasks for retrieval** (i.e., multi-pathway architecture and learning-to-rank tasks).

## 2.1 OVERVIEW OF ONLINE ADVERTISING SYSTEM

Figure 1 illustrates a typical cascade ranking system for online advertising systems. The system is comprised of four main stages: Matching, Pre-ranking, Ranking, and Mix-ranking. The "Matching" and "Pre-ranking" stages serve as retrieval mechanisms, focusing on selecting candidate ads without directly impacting their billing. Conversely, the "Ranking" stage plays a dual role by determining both the selection and billing of ads. It achieves this by predicting the eCPM (Effective Cost Per Mille) of each ad, serving as the standard for billing after its exposure. The "Mix-ranking" stage integrates the outputs from the advertising and recommendation systems to decide the final set of items presented to the user. For advertising systems, the differences in the goal of the retrieval and ranking stages lead to some technical differences, such as training objectives, system design, evaluation metrics, etc. In this work, **we focus on the scaling laws of retrieval stages**.

Figure 2 shows the typical algorithm architecture of the Matching and Pre-ranking stages in advertising systems. They all adopt a **multi-pathway ensemble architecture**. The Matching stage can be divided into model-based pathways and rule-based pathways. The rule-based pathway can quickly filter out irrelevant ads using predefined criteria, ensuring compliance with regulations and improving ad relevance, while providing flexibility and transparency in the ad selection process. The model-based pathway uses machine learning algorithms to predict and select ads, achieving higher performance ceilings in ad selection. The Pre-ranking stage operates on the set of ads already selected by the Matching stage, and thus only employs the model-based pathways.

Model-based pathways in the Matching and Pre-ranking stages can typically be categorized based on different modeling objectives, such as revenue-oriented, user-interest-oriented, and lifetime value-oriented pathways. In advertising systems, revenue-oriented pathways usually carry the most weight. Since the earlier stages only need to perform set selection, advertising systems often employ Learning-to-rank methods to learn the optimal ranking order to maximize the revenue objective. The Learning-to-rank pathway is the most prevalent and important in the system, as it is typically the primary focus for algorithm researchers to iterate and improve (Wang et al., 2018; Jagerman et al., 2022; Zheng et al., 2024; Lyu et al., 2023; Wang et al., 2024; 2025). Therefore, **we use the Learning-to-rank pathway as a case study to verify scaling laws** in online advertisement retrieval.

## 2.2 FORMULATION OF RETRIEVAL TASKS

Here, we give the formulation of retrieval tasks represented by learning-to-rank. Following Zheng et al.; Wang et al.(2024; 2025), we randomly draw samples from the Matching, Pre-ranking, Ranking, and Mix-ranking stages. We use all the samples for training the model of the Matching stage, and we use the samples from all stages except the Matching stage to train the Pre-ranking models. The **training set** can be formulated as:

$$D_{train} = \{(f_{u_i}, \{f_{a_i^j}, v_i^j, ecpm_i^j | 1 \le j \le n\})_i\}_{i=1}^{N} \tag{1}$$

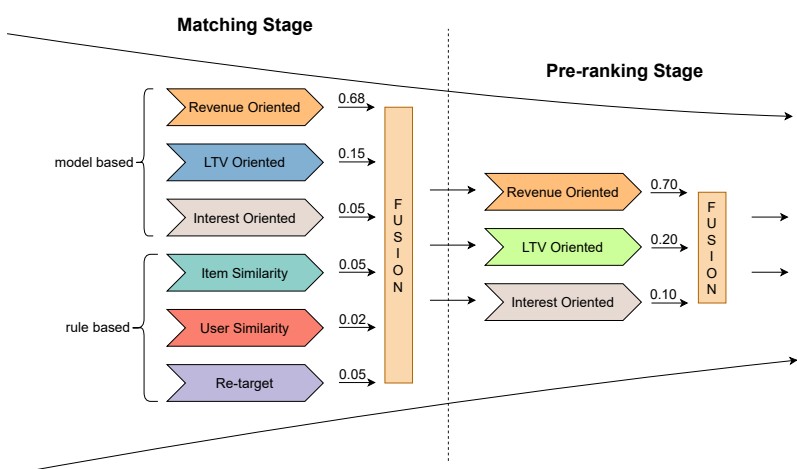

Figure 2: A typical multi-pathway architecture of Matching and Pre-ranking stages. Each pathway has its own weight, determining the proportion of ads sent to the next stage after the Fusion module.

where $u_i$ means the user of the $i$-th impression in the training set, $a^j$ means the $j$-th material for ranking in the system. $N$ is the number of impressions of $D_{train}$. The $i$ in $a_i^j$ means that the sample $a^j$ corresponds to impression $i$. The size of the materials for each impression is $n$. $f_{(\cdot)}$ means the feature of $(\cdot)$. $v_i^j$ is the ground truth rank index (the higher the better) of the pair $(u_i, a^j)$. The rank index is the relative position of the sampled ads within the system queue, ordered by their original positions. Formally, this can be formulated as $\mathbf{1}(v_i > v_j) = \mathbf{1}(s_i > s_j) \vee (\mathbf{1}(s_i = s_j) \wedge \mathbf{1}(\text{score}_i > \text{score}_j))$ where $s$ denotes the sampling stage of the ad, and $score$ represents the system-assigned score within that stage. $ecpm_i^j$ **is the eCPM (Effective Cost Per Mille) predicted by the Ranking stage, which can be regarded as the expected revenue for the exposure of** $ad_i^j$. If $ad_i^j$ does not win in the Matching or Pre-ranking and thus does not enter the Ranking stage, the $ecpm_i^j$ equals 0. **The eCPM information in the dataset is only used to construct our offline evaluation metric**.

The **learning objective** in Learning-to-Rank is to learn the order of all or part of the training set $D_{train}$. This can typically be achieved using different types of methods, such as point-wise (Li et al., 2007), pair-wise (Burges et al., 2005), and list-wise (Wang et al., 2018; 2024; Zheng et al., 2024) approaches. We employ ARF (Wang et al., 2024) to train the models in our scenario.

## 3 SCALING LAWS OF BUDGET AND REVENUE

In this section, **we present a lightweight paradigm for identifying the scaling laws of computation budget (namely, machine cost) and revenue**. We also verify whether MLP (mlp, 1958), DSSM (Huang et al., 2013), and Transformer (Vaswani et al., 2017) exhibit scaling laws in the context of online advertisement retrieval. In sub-section 3.1, we first introduce $R/R^*$, a carefully designed offline metric that is highly linearly correlated with online revenue. In sub-section 3.2, we then demonstrate that DSSMs, MLPs, and Transformers exhibit a broken neural scaling law (Caballero et al., 2023) between FLOPs and the offline metric $R/R^*$. In sub-section 3.3, we finally propose a simulation algorithm to estimate machine cost based on model size parameters. Leveraging the $R/R^*$ metric, the observed scaling laws, and the cost simulation algorithm, we are able to reliably estimate online machine costs and revenue gains from offline model configurations.

### 3.1 EFFECTIVE SURROGATE METRIC FOR ONLINE REVENUE

Traditional offline metrics (e.g., OPA (Wang et al., 2024), Recall, and NDCG) used for online advertising retrieval often ignore the revenue difference of the ground-truth ads and treat them equally. This naturally creates a gap between offline metrics and online revenue. To mitigate this gap, we propose a novel metric named $R/R^*$ based on $D_{train}$. The $R/R_{(i)}^*$ for impression $i$ in $D_{train}$ can be formulated as Eq 2:

$$R/R^*_{(i)}(m) = \frac{\sum(\mathcal{P}^{\downarrow}_{\mathcal{M}^{(i,\cdot)}}[:m;:] * \mathbf{ecpm}_i^T)}{\sum(\mathcal{P}^{\downarrow}_{\mathbf{v}_i}[:m;:] * \mathbf{ecpm}_i^T)} \tag{2}$$

where $\mathcal{P}^{\downarrow}_{\mathcal{M}^{(i,\cdot)}}$ and $\mathcal{P}^{\downarrow}_{\mathbf{v}_i}$ denote the hard permutation matrices (explained in Appendix B.1) sorted by $\mathcal{M}^{(i,\cdot)}$ and $\mathbf{v}_i$ respectively. $\mathcal{M}^{(i,\cdot)}$ denotes the prediction vector of model $\mathcal{M}$ for the $i$-th impression of $D_{train}$. When the input vector length is $n$, the hard permutation matrix is an $n \times n$ square matrix. The notation $[:m;:]$ denotes the operation of taking the first $m$ rows of the matrix. $\mathbf{ecpm}_i$ is a row vector that represents the expected revenue of each ad of the $i$-th impression. $R/R^*$ is the average of $R/R^*_{(i)}$ on $D_{train}$. $m$ is a hyper-parameter of $R/R^*$.

$R/R^*$ aligns better with online revenue mainly because it explicitly considers the commercial value (eCPM) of each ad, directly reflects the goal of maximizing revenue, and reduces the gap between offline evaluation and online performance. **The following theorem establishes the theoretical connection between $R/R^*$ and online revenue under a set of plausible assumptions. The proof is provided in Appendix B.2.**

**Theorem 3.1** (Linearity between $R/R^*$ and Online Revenue). *Under Assumptions 1-4, the offline metric $R/R^*(m)$ is linearly correlated with the online revenue. Specifically, there exists a positive constant $\gamma > 0$ such that for a certain pathway, the online revenue satisfies:*

$$\text{Online Revenue} = \gamma \cdot R/R^*(m) + \text{constant}.$$

**Assumption 1.** *The training data $D_{train}$ and online data are independent and identically distributed.*

**Assumption 2.** *The improvement in $R/R^*$ for a single pathway is proportional to the improvement in $R/R^*$ of the entire stage ensemble's output set. It can be formulated as:*

$$\Delta R/R^*_{ensemble}(m) = \alpha \Delta R/R^*_{single}(m) \tag{3}$$

*where $R/R^*_{single}$ are $R/R^*_{ensemble}$ the $R/R^*$ for single pathway and entire stage, respectively. $\alpha$ is a constant.*

**Assumption 3.** *Only the top-$m$ ads from the retrieval stages will be selected by the post-stages for exposure; the exposure probability of $ad_i^j$ is $p_i^j$. We define $\beta_i^j = \frac{eCPM_i^j}{\sum(\mathcal{P}^{\downarrow}_{\mathcal{M}^{(i,\cdot)}}[:m;:] * \mathbf{ecpm}_i^T)}$ as the normalized contribution of the $j$-th ad (on the top-$m$ sets sorted by retrieval stages) of the $i$-th impression. We assume that $\beta_i^j$ and $p_i^j$ are invariant across different impressions, depending only on the position $j$.*

**Assumption 4.** *The ranking stage predicts CPMs that are consistent with the true per-impression revenue. Moreover, for each impression $i$, the sum of the top-$M$ predicted CPMs (denoted as $\mathcal{P}^{\downarrow}_{\mathbf{v}_i}[:m;:] * \mathbf{ecpm}^T$) is statistically independent of the retrieval model's performance metric $R/R^*_{(i)}$.*

In a mature advertising system, we think Assumptions 1, 3 and 4 holds approximately true. Assumption 2 also holds approximately true for pathways with dominant weights. Therefore, the $R/R^*$ of the Learning-to-rank pathway (with dominant weight) and the revenue of online advertising systems should be approximately linearly related. To verify the effectiveness of $R/R^*$, we also conduct extensive online experiments based on our A/B test platform. We deploy several different models to empirically study the relationship between online revenue and offline metrics, which differ in feature engineering, surrogate loss, model structure, and model size. Table 1 shows the experimental results. **It is evident that $R/R^*$ exhibits the strongest linear correlation with online revenue, with an $R^2$ of 0.902, indicating that $R/R^*$ is a more advanced offline metric**. Note that $m$ is 2 in the experiment of Table 1. Due to space limitations, more experimental details are shown in Appendix Sections B.3 and B.4.

## 3.2 SCALING LAWS OF FLOPs AND OFFLINE METRIC

In this section, we investigate whether there exist scaling laws for different model architectures between computational effort and online revenue under the setting described in Section 2. Thanks to the strong linear correlation between $R/R^*$ and online revenue, we can transform the identification

Table 1: Linear correlation analysis between online revenue and four offline ranking metrics across two stages (Pre-ranking and Matching). $R^2$ measures the goodness of fit (higher is better), while average deviation (Avg. Dev) and maximum deviation (Max. Dev) reflect the fitting errors (lower is better). Best results are bolded.

| Scenario | Correlation Measures | Offline Metrics | | | |
|---|---|---|---|---|---|
| | | $R/R^*$ | NDCG | Recall | OPA |
| Pre-ranking | $R^2 \uparrow$ | **0.902** | 0.793 | 0.509 | 0.140 |
| | Avg. Dev$\downarrow$ | **0.243** | 0.311 | 0.534 | 0.634 |
| | Max. Dev$\downarrow$ | **0.483** | 0.819 | 1.139 | 1.690 |
| Matching | $R^2 \uparrow$ | **0.725** | 0.315 | 0.504 | 0.259 |
| | Avg. Dev$\downarrow$ | **0.287** | 0.626 | 0.452 | 0.558 |
| | Max. Dev$\downarrow$ | **1.082** | 1.712 | 1.100 | 1.515 |

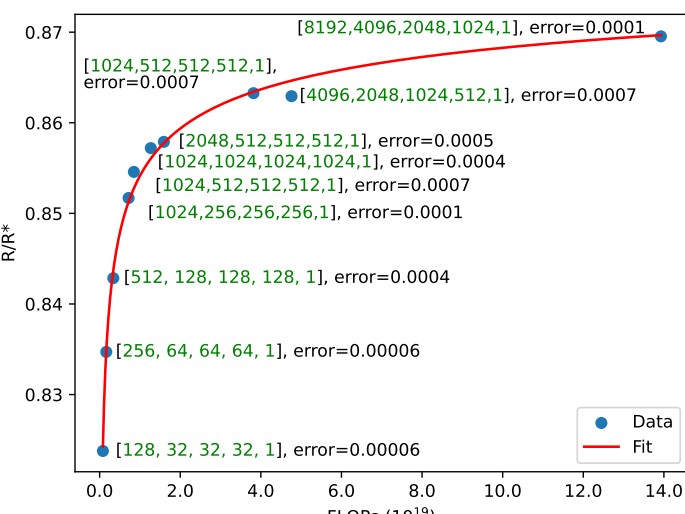

Figure 3: Scaling Performance regarding the FLOPs and $R/R^*$ of MLP models under $Scene_1$. The size of the MLP network and the deviation between the fitted curve and the true values (referred to as the "error") are annotated in the figure. The $R^2$ value of the curve fitting is 0.996. [1024, 256, 256, 256, 1] indicates an MLP model with 5 layers, where the output sizes are 1024, 256, 256, 256, and 1, respectively. The total dimensions of input features for all models are 3328.

of the scaling laws from costly online A/B tests to efficient offline experiments. We train multiple models with different model sizes offline, and then collected their converged $R/R^*$ metrics and calculated the FLOPs for each model **to examine the existence of scaling laws for mainstream model architectures (i.e., Transformer (Vaswani et al., 2017), MLP (mlp, 1958), and DSSM (Huang et al., 2013))** under two different scenarios (denoted as $Scene_1$ and $Scene_2$) in our system. Specifically, we examine whether the collected data follows the Broken Neural Scaling Law (BNSL), a statistical framework proposed by Caballero et al.(2023) and empirically validated across numerous deep learning tasks in computer vision and natural language processing. BNSL is formulated as shown in Eq 4:

$$R/R^* = a + (b * FLOPs^{-c_0}) \prod_{i=1}^{t} (1 + (\frac{FLOPs}{d_i})^{1/f_i})^{(-c_i * f_i)} \qquad (4)$$

where $a, b, c_0, c_i, f_i$ are learnable parameters and $t$ is hyperparameter of BNSL. $t$ characterizes the degrees of freedom of BNSL, and in practice, we typically take it as 6. The scaling factor is the FLOPs, and the dependent variable is our proposed offline metric $R/R^*$. We use basin-hopping (Wales and Doye, 1997) to fit the BNSL.

Due to space limitations, we present the results and analysis for MLP models in $Scene_1$ as a representative example in the main text. The scaling laws for MLPs and DSSMs are established within a structured design space defined by near-proportional scaling, where the ratio between consecu-

tive hidden layer sizes does not exceed 20. This constrained strategy reflects a common industrial practice for expanding proven base models. Arbitrarily scaled MLPs and DSSMs may not share the same scaling laws, as discussed in Appendix F.1. Figure 3 illustrates the fitted BNSL curve, along with the original data points. The x-axis represents the scaling factor (FLOPs), and the y-axis represents the dependent variable ($R/R^*$). Each data point corresponds to a model of a different size, with the number of layers and units per layer annotated in the figure. The $R^2$ value of the curve fitting is 0.996, indicating a strong correlation between the predicted values and the observed data points. **This provides compelling empirical evidence that a Broken Neural Scaling Law governs the scaling behavior of MLP models** in the typical online advertisement retrieval scenario (see Section 2). We observe consistent scaling behaviors across other scenarios and architectures. Such a scaling law enables us to predict model performance under different computational budgets and better understand efficiency-performance trade-offs. **Comprehensive experimental setups and results for Transformer, DSSM, and MLP across both $Scene_1$ and $Scene_2$ are provided in Appendix C**, along with a discussion on scaling laws in NLP and online advertising systems.

Thanks to the strong linear relationship between $R/R^*$ and online revenue, we can establish a prediction from FLOPs to online revenue. To further validate whether the accuracy of the FLOPs-to-revenue estimation meets the needs of model development and iteration, we deploy two MLP models for 7 days, each with 10% of online traffic. The FLOPs for these models are $1.26 \times 10^{19}$ and $2.29 \times 10^{19}$, respectively. Based on the scaling law, the estimated online revenue gains are 0.33% and 0.62%, while the actual online revenue increases are 0.38% and 0.69%, respectively. In our scenario, this accuracy is more than sufficient for guiding model development and iteration. The MLP network has the same size configuration [1024, 512, 512, 512, 1] for both models, while the dimensions of the input features are 5128 and 10128, respectively. That is, the scaling law—originally learned by scaling model width under fixed input size—remains accurate even when model scale is increased through dimension expansion of input features, highlighting the robustness and generalization of the FLOPs-to-revenue estimation. Notably, the estimation errors of these two models are smaller than the average error of figure 4a. This may be because the models in figure 4a were deployed for one day, while these two models were deployed for 7 days, which means the observations of figure 4a would have higher variance. This observation also suggests the effectiveness and robustness of our proposed offline metric $R/R^*$.

### 3.3 Mapping Model Settings to Machine Cost

For real-world model deployment, **we need the scaling law of revenue and computation cost (i.e., machine cost)**, rather than the relationship of revenue and FLOPs. Our ultimate goal is to build an end-to-end offline framework that maps model configurations to both revenue and machine cost, enabling ROI-aware model design without online A/B testing. The revenue side of this mapping is addressed in Sections 3.1 and 3.2: we establish a robust scaling law from **Model** $\rightarrow$ **FLOPs** $\rightarrow$ $R/R^*$ $\rightarrow$ **Revenue**, where FLOPs are analytically computed from model architecture (e.g., using TensorFlow's `tf.profiler`). However, the **cost side** cannot be similarly bridged via FLOPs: actual machine cost depends heavily on environment-specific factors—such as hardware, software stack, and system optimizations—making the FLOPs-to-cost relationship highly non-linear and unpredictable across platforms. Thus, a direct **FLOPs** $\rightarrow$ **Machine Cost** mapping is infeasible for general use.

To overcome this, we introduce **model size parameters** (e.g., layer dimensions) as a more stable intermediate proxy. While FLOPs and model size are closely related, the latter better captures the structural footprint that system-level machine cost estimators rely on. Hence, we shift to a **Model** $\rightarrow$ **Size Parameters** $\rightarrow$ **Machine Cost** pathway, where the key challenge is estimating cost from size *without online deployment*. We discuss two paradigms for this mapping: (1) expert-driven estimation based on system optimization experience, and (2) offline simulation-based estimation.

The first approach relies on domain experts to empirically correlate model size with machine cost. For example, Fang et al.(2024) estimated machine costs for standard Transformer models on A100 GPUs using PyTorch. However, in industrial advertising systems, custom training/serving frameworks (e.g., first-layer optimizations in Section D.1) and heterogeneous machine configurations introduce complex, non-standardized dependencies, limiting the generalizability and accuracy of expert-based estimation.

The second approach—**offline simulation**—offers a more systematic, reproducible, and scalable solution. Given a model architecture and its **size parameters**, we can generate its meta file offline. As detailed in Algorithm 1 (Appendix D.2), our Machine Cost Estimation Tool (MCET) takes such meta files as input and simulates the computational footprint to estimate the required number of machines. Using a single GPU, MCET can estimate the machine count for a model within 30 minutes, without actual deployment. By combining this with unit machine pricing, we obtain accurate cost projections. Notably, for training cost estimation, synthetic labels are used to complete the forward/backward pass simulation. This approach enables rapid, reliable machine cost evaluation during model iteration, significantly reducing both time and risk in deployment planning. In our production environment, the machine cost estimated by MCET is consistently within 5% of the actual cost, a level of accuracy fully sufficient for industrial resource planning and ROI calculations.

# 4 APPLICATIONS IN MODEL DESIGNING AND RESOURCE ALLOCATION

In this section, we demonstrate two real-world applications of the scaling laws established in Section 3: **ROI-constrained model design** and **multi-scenario resource allocation**. These applications enable cost-aware, data-driven model development without extensive online testing.

To facilitate exposition, we define the following notations:
- $\mathcal{SP}$: The size parameters of a model. For MLPs and DSSMs, $\mathcal{SP} = [a_0, a_1, \ldots, a_n]$ specifies layer dimensions (e.g., input/output sizes). For Transformer-based models, $\mathcal{SP}$ includes embedding dimension, number of layers, attention heads, and feed-forward network (FFN) size. $meta_{\mathcal{SP}}$ denotes the meta file generated from the model configured with $\mathcal{SP}$.
- $F(\mathcal{SP})$: The FLOPs of the model with $\mathcal{SP}$, computed analytically or via profiling tools.
- $BNSL$: The mapping from FLOPs to $R/R^*$, following the form in Equation 4.
- $G$: The linear mapping from $R/R^*$ to online revenue, calibrated from historical A/B tests.
- $MCET$: Our Machine Cost Estimation Tool, which takes $meta_{\mathcal{SP}}$ as input and simulates system-level resource demands to estimate machine cost, denoted as $MCET(meta_{\mathcal{SP}})$.

The functions $G$, $BNSL$, $F$, and $MCET$ are derived from Sections 3.1, 3.2, and 3.3, respectively.

## 4.1 ROI-CONSTRAINED MODEL DESIGNING

Industrial systems require that any model deployment achieves a minimum return on investment (ROI) threshold $\lambda$, typically set by business stakeholders. We formalize this as:

$$ROI = \frac{G(BNSL(F(\mathcal{SP})))}{MCET(\text{meta}_{\mathcal{SP}})} \geq \lambda \tag{5}$$

Our goal is to maximize revenue under this constraint:

$$\begin{aligned} \max_{\mathcal{SP}} \quad & G(BNSL(F(\mathcal{SP}))) \\ \text{s.t.} \quad & \frac{G(BNSL(F(\mathcal{SP})))}{MCET(\text{meta}_{\mathcal{SP}})} \geq \lambda \end{aligned} \tag{6}$$

While $G \circ BNSL \circ F$ is monotonically increasing in $\mathcal{SP}$, $MCET$ is non-monotonic due to system-level effects (e.g., kernel fusion, memory access patterns), making the problem non-convex. Analytical solutions or binary search are infeasible.

We solve Eq. 6 via grid search over plausible $\mathcal{SP}$ configurations. MCET evaluations are the most time-consuming part, but are still feasible to run offline for grid search. Using 10 GPU machines, we evaluate about 1,000 configurations in about two days (details in Appendix D.3).

**Case Study 1: Pre-ranking Model Optimization.** For the MLP-based pre-ranking model, we identify the optimal $\mathcal{SP}^* = [16128, 1024, 512, 512, 512, 1]$ under our threshold $\lambda^*$, improving over the baseline $[3328, 1024, 256, 256, 256, 1]$. This yields a **+0.85% online revenue gain**. Such an improvement is considered significant in our advertising scenario.

**Case Study 2: Matching Pathway Optimization.** Guided by scaling laws for Transformer models, we introduced a lightweight Transformer-based pathway in the Matching stage, under the constraint of fixed machine cost and $ROI > \lambda^*$. This change led to a **+1.0% increase in overall revenue**.

This approach replaces costly trial-and-error with systematic, offline optimization, enabling rapid iteration on model designs. Without the scaling law, it would be nearly impossible to test the ROI of hundreds of model configurations through online deployment. The time and resource costs of deploying so many models would far outweigh the incremental revenue from the optimized model derived using the scaling law.

## 4.2 MULTI-SCENARIO RESOURCE ALLOCATION

Industrial online advertising systems often deploy different models across scenarios—such as matching, pre-ranking, and ad placements—each with distinct traffic patterns and business objectives. To maximize overall revenue under a fixed total machine budget, we propose a scaling-law-driven framework for multi-scenario resource allocation, ensuring $ROI \geq 1$ per scenario.

Let there be $t$ scenarios with a shared machine budget $B$. Our goal is to allocate resources across scenarios to maximize total revenue without exceeding $B$, while maintaining $ROI \geq 1$ in each. This leads to the optimization problem in Eq. 7:

$$\max_{\{\mathcal{SP}_i\}_{i=0}^{t-1}} \sum_{i=0}^{t-1} G\big(BNSL(F(\mathcal{SP}_i))\big)$$

$$\text{s.t.} \sum_{i=0}^{t-1} MCET(\text{meta}_{\mathcal{SP}_i}) \leq B \tag{7}$$

$$\frac{G\big(BNSL(F(\mathcal{SP}_i))\big)}{MCET(\text{meta}_{\mathcal{SP}_i})} \geq 1, \quad \forall i \in \{0, \ldots, t-1\}$$

where $\mathcal{SP}_i$ denotes the model configuration (e.g., layer sizes) in scenario $i$, and $MCET(\cdot)$ estimates machine cost from the model's meta file. This problem can be solved using the same grid search approach as described in sub-section 4.1. The computational complexity scales linearly with $t$ compared to the single-scenario case (Eq. 8), dominated by MCET evaluations.

**Case Study 3: Matching vs. Pre-ranking.** We treat these two stages as separate scenarios ($t = 2$). Despite the initially suboptimal allocation, our method achieves a **+2.8% online revenue gain** under the same machine budget, with all scenarios satisfying $ROI \geq 1$.

**Case Study 4: Two Heterogeneous Ad Placements.** We further validate our framework on two distinct ad placements (i.e., $Scene_1$ vs. $Scene_2$) with different user engagement patterns. Applying the same optimization, we observe a **+0.45% increase in total revenue**, demonstrating the generalizability of our approach across diverse scenario types.

For large $t$, exhaustive grid search becomes infeasible. In such cases, we can employ heuristic search (e.g., Bayesian optimization) or restrict scaling to proportional changes of a base model. This framework enables automated, data-driven resource allocation across complex multi-scenario systems, reducing reliance on manual tuning. Further details are provided in Appendix D.4.

## 5 CONCLUSION

We have presented a systematic study of scaling laws in online advertisement retrieval systems, addressing the high cost and inefficiency of online experimentation in industrial settings. By introducing an offline metric $R/R^*$ and a machine cost estimation method, we propose a lightweight paradigm that transforms costly online trials into efficient offline analyses. This paradigm enables accurate modeling of the relationship between machine cost and online revenue—mediated by model configurations—across diverse model architectures. We also conduct extensive experiments to validate the existence of scaling laws under different model architectures (i.e., MLP, DSSM, Transformer) and scenarios in an online advertisement retrieval system. With our proposed paradigm, we demonstrate practical applications in ROI-constrained model design and multi-scenario resource allocation, achieving a substantial 5.1% improvement in online revenue, validated via A/B testing. The framework supports data-driven decision-making and accelerates model iteration in production environments. Limitations and future directions are discussed in Appendix F.

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

# A    RELATED WORK

## A.1    NEURAL SCALING LAWS

The neural scaling laws, widely recognized in Natural Language Processing and Computer Vision areas, establish predictable relationships between model performance and key factors such as model size, dataset size, computational cost, and post-training error rates. Hestness et al.(2017) introduced a unified scaling law applicable to machine translation, image classification, and speech recognition, noting that different tasks exhibit distinct scaling coefficients. Kaplan et al.(2020) further elaborated on these laws, defining them through four parameters: model size (number of parameters), dataset size (number of tokens), computational cost (FLOPs), and training loss (perplexity per token). These relationships were empirically validated, including during the training of GPT-3 (Brown, 2020). Subsequently, Hoffmann et al.(2022) presented the Chinchilla Scaling Law, which differs somewhat from (Kaplan et al., 2020) because of their different training setups. In addition to estimating the training loss of the model, Isik et al.(2024) further verified that there also exist scaling laws between the downstream task metrics and the model parameters. Caballero et al.(2023) found that many scaling behaviors of artificial neural networks follow a smoothly broken power law (BNSL), verified on various NLP and CV tasks, covering a wide range of downstream tasks and model structures.

In the recommendation area, Shin et al.(2023) and Zhang et al.(2024) studied on whether there exist scaling laws for recommendation models and primarily provided qualitative conclusions. Fang et al.(2024) first proposed a quantitative scaling law in the recommendation area, which describes the relationship between the amount of training data, the size of model parameters, and an offline metric for query-document retrieval. Based on this scaling law, the optimal amount of data and model parameter allocation can be solved under a given total training resource. However, obtaining a multivariate scaling law requires a large number of experiments, and the offline metrics might not be a good indicator for online metrics, these make it somewhat difficult to apply in real-world industrial applications. In this paper, we focus on how to obtain the scaling law function between the model's scalable hyper-parameters (such as the FLOPs) and the online revenue (the primary goal of the advertising system) with only a small amount of experimental cost.

## A.2    MODELS AND EVALUATION FOR ONLINE ADVERTISEMENT RETRIEVAL

Online advertising systems often adopt a cascade ranking framework (Wang et al., 2011; Chen et al., 2017; Gallagher et al., 2019; Qin et al., 2022; Wang et al., 2025). The cascade ranking usually includes two types of stages, namely retrieval and ranking. The retrieval stages take a set of terms as input and supply the top-k predicted terms to the next stage, which mainly focuses on order accuracy. The ranking stages in advertising systems should focus on not only the order accuracy but also the calibration accuracy (Huang et al., 2022; Sheng et al., 2023; Zhao et al., 2024; Dai et al., 2025). These lead to some technique differences in the retrieval and ranking stages, such as optimization objectives, surrogate losses, and design of evaluation metrics. In this work, we focus on the retrieval stages of online advertising systems.

The retrieval stages select the top-k set for the next stage, typically including multiple ranking pathways. In the retrieval stages, which normally refers to the "Matching" or "Pre-rank" stage in industrial systems, the models often adopt a twin-tower (Huang et al., 2013; Covington et al., 2016) or MLP (mlp, 1958; Hornik et al., 1989; Zhu et al., 2018; Wang et al., 2020; 2024; 2025) architecture. Some systems may also adopt more complex architectures like transformer and its variants (Vaswani et al., 2017; Pancha et al., 2022; Rajput et al., 2023; Zhai et al., 2024; Han et al., 2025; Deng et al., 2025; Zhou et al., 2025a;b). In terms of objective design, industrial systems usually aim to estimate CTR (He et al., 2014; Zhou et al., 2018; 2019) and exposure probability of ads, and use learning-to-rank methods (Burges et al., 2005; Wang et al., 2018; Jagerman et al., 2022; Wang et al., 2024; 2025) to learn the results of ranking models. For evaluation of retrieval tasks, researchers commonly use OPA, NDCG, and Recall (Swezey et al., 2021; Zangerle and Bauer, 2022; Bauer et al., 2024; Wang et al., 2024; 2025). However, existing metrics often have complex relationships with online revenue; thus, we propose $R/R^*$, which exhibits a strong linear relationship with online revenue, to reduce the experimental cost of identifying the scaling laws for industry scenarios.

# B    MORE EXPLANATIONS, ANALYSIS AND EXPERIMENT DETAILS OF THE OFFLINE METRIC $R/R^*$

## B.1    EXPLANATIONS OF THE CONCEPT OF PERMUTATION MATRIX

The concept of a *hard permutation matrix* is central to the formulation of our $R/R^*$ metric and is widely used in differentiable sorting literature (Grover et al. (2019); Prillo and Eisenschlos (2020); Petersen et al. (2021)). A permutation matrix is a square matrix that has exactly one entry of 1 in each row and each column, and 0s elsewhere. When multiplied by a vector, it reorders the elements of that vector.

Formally, for an input vector $\mathbf{a} = [a_1, a_2, \ldots, a_n]$, let $\mathbf{b} = [b_1, b_2, \ldots, b_n]$ be its sorted version (e.g., in descending order: $b_1 \geq b_2 \geq \ldots \geq b_n$). The hard permutation matrix $P$ that maps $\mathbf{a}$ to $\mathbf{b}$ is defined such that its element $P_{i,j} = 1$ if and only if the element $a_j$ from the original vector is placed at position $i$ in the sorted vector $\mathbf{b}$. This satisfies the equation $\mathbf{b} = P\mathbf{a}$.

**Example:** Consider a vector $\mathbf{a} = [2, 4, 1, 3]$. Its sorted version in descending order is $\mathbf{b} = [4, 3, 2, 1]$. The hard permutation matrix $P$ that satisfies $\mathbf{b} = P\mathbf{a}$ is constructed as follows:

- Row 1 (top position in $\mathbf{b}$, value 4): the value 4 was originally at index 2 in $\mathbf{a}$, so $P_{1,2} = 1$.
- Row 2 (value 3): the value 3 was originally at index 4 in $\mathbf{a}$, so $P_{2,4} = 1$.
- Row 3 (value 2): the value 2 was originally at index 1 in $\mathbf{a}$, so $P_{3,1} = 1$.
- Row 4 (value 1): the value 1 was originally at index 3 in $\mathbf{a}$, so $P_{4,3} = 1$.

Thus, the matrix is:

$$P = \begin{bmatrix} 0 & 1 & 0 & 0 \\ 0 & 0 & 0 & 1 \\ 1 & 0 & 0 & 0 \\ 0 & 0 & 1 & 0 \end{bmatrix}$$

Multiplying $P$ by $\mathbf{a}$ verifies the result: $P\mathbf{a} = [4, 3, 2, 1]^T = \mathbf{b}$.

In the context of $R/R^*$, the matrices $\mathcal{P}_{\mathcal{M}(i,:)}^{\downarrow}$ and $\mathcal{P}_{\mathbf{v}_i}^{\downarrow}$ in Eq. (2) are precisely such hard permutation matrices, which reorder the ads based on the model's predictions and the ground-truth ranking, respectively.

## B.2    PROOF OF THEOREM 3.1

This appendix provides the formal proof for **Theorem 3.1** stated in Section 3.1. We restate the theorem for clarity:

**Theorem 1 (Linearity between $R/R^*$ and Online Revenue).** Under Assumptions 1-4, the offline metric $R/R^*(m)$ is linearly correlated with the online revenue. Specifically, there exists a positive constant $\gamma > 0$ such that for a certain pathway, the online revenue satisfies:

$$\text{Online Revenue} = \gamma \cdot R/R^*(m) + \text{constant}.$$

*Proof.* The proof is structured into three parts, establishing the linear relationship from the ensemble model down to a single pathway and finally generalizing to the online setting.

**Part 1: Linearity for the Ensemble Model on $D_{\text{train}}$.**

Let $p_i^j$ denote the exposure probability of ad $a_i^j$. The total revenue on the training data is:

$$\text{Revenue}(D_{\text{train}}) = \sum_{i=1}^{N} \sum_{j=1}^{n} p_i^j \cdot \text{eCPM}_i^j. \tag{8}$$

By Assumption 3, the normalized contribution $\beta_i^j = \frac{\text{eCPM}_i^j}{\sum(\mathcal{P}_{\mathcal{M}(i,\cdot)}^{\downarrow}[:m,:] \cdot \mathbf{ecpm}_i^T)}$ and the exposure probability $p_i^j$ depend only on the position $j$, denoted as $\beta^j$ and $p^j$ respectively. Let $C_i = \sum(\mathcal{P}_{\mathbf{v}_i}^{\downarrow}[:$

$m, :] \cdot \mathbf{ecpm}_i^T$), which is the top-$m$ eCPM sum under the ground-truth ranking. We can rewrite the revenue as:

$$
\begin{aligned}
\text{Revenue}(D_{\text{train}}) &= \sum_{i=1}^{N} C_i \cdot \left( \sum_{j=1}^{n} p^j \beta^j \cdot \left( \frac{1}{C_i} \sum_{j=1}^{m} \mathcal{P}_{\mathcal{M}^{(i,\cdot)}}[: m, :] \cdot \mathbf{eCPM}_i^T \right) \right) \\
&= \left( \sum_{i=1}^{N} C_i \cdot R/R_i^*(m) \right) \cdot \left( \sum_{j=1}^{n} p^j \beta^j \right),
\end{aligned}
\tag{9}
$$

where the term $\sum_{j=1}^{m} \mathcal{P}_{\mathcal{M}^{(i,\cdot)}}[: m, :] \cdot \mathbf{eCPM}_i^T$ is the numerator of $R/R_i^*(m)$, and $C_i$ is its denominator.

Under Assumption 4, $C_i$ is statistically independent of $R/R_i^*(m)$. Therefore, the revenue simplifies to:

$$
\begin{aligned}
\text{Revenue}(D_{\text{train}}) &= \left( \sum_{i=1}^{N} C_i \right) \cdot \left( \frac{1}{N} \sum_{i=1}^{N} R/R_i^*(m) \right) \cdot \left( \sum_{j=1}^{n} p^j \beta^j \right) \\
&= N \cdot \mathbb{E}(C) \cdot R/R^*(m) \cdot \left( \sum_{j=1}^{n} p^j \beta^j \right)
\end{aligned}
\tag{10}
$$

where $R/R^*(m) = \frac{1}{N} \sum_{i=1}^{N} R/R_i^*(m)$, and $\mathbb{E}(C) = \frac{1}{N} \sum_{i=1}^{N} C_i$. Defining the constant $\gamma_1 = N \cdot \mathbb{E}(C) \cdot \sum_{j=1}^{n} p^j \beta^j$, we obtain the linear relationship:

$$
\text{Revenue}(D_{\text{train}}) = \gamma_1 \cdot R/R_{\text{ensemble}}^*(m).
\tag{11}
$$

**Part 2: Extension to a Single Dominant Pathway.**

From Part 1, we have $\text{Revenue} = \gamma_1 \cdot R/R_{\text{ensemble}}^*(m)$. Consider a change $\Delta R/R_{\text{single},k}^*(m)$ in a single pathway $k$, holding others constant. By Assumption 2 for a dominant pathway, the change in the ensemble metric is:

$$
\Delta R/R_{\text{ensemble}}^*(m) = \alpha \cdot \Delta R/R_{\text{single},k}^*(m).
\tag{12}
$$

The corresponding change in revenue is:

$$
\Delta\text{Revenue} = \gamma_1 \cdot \Delta R/R_{\text{ensemble}}^*(m) = \gamma_1 \alpha \cdot \Delta R/R_{\text{single},k}^*(m).
\tag{13}
$$

Letting $\gamma_2 = \gamma_1 \alpha$, we have $\Delta\text{Revenue} = \gamma_2 \cdot \Delta R/R_{\text{single},k}^*(m)$, establishing the proportional relationship for a dominant pathway.

**Part 3: Generalization to Online Revenue.**

By Assumption 1, the online data and $D_{\text{train}}$ are independent and identically distributed (i.i.d.). Therefore, the linear relationship derived on $D_{\text{train}}$ holds for the online revenue as well. Thus, for a dominant pathway, there exists a constant $\gamma > 0$ such that:

$$
\Delta\text{Online Revenue} = \gamma \cdot \Delta R/R_{\text{single}}^*(m),
\tag{14}
$$

which demonstrates the Linearity between online revenue and $R/R^*$ under Assumptions 1 to 4, namely completes the proof of Theorem 3.1. $\qquad\square$

### B.3  MORE EXPERIMENT DETAILS AND THE VISUALIZATION OF THE CORRELATION BETWEEN ONLINE REVENUE AND OFFLINE METRICS

Due to the limitations on experimental traffic, we can only run up to 5 A/B test groups simultaneously, which means some experimental groups were launched on different days. The online experiments corresponding to each model use at least 5% of the traffic. We collect each model's online revenue from the A/B test platform and offline metrics from the training log.

In the main text, we only show some correlation measures (e.g., $R^2$) of the linear correlation between online revenue and offline metrics. Here we give the visualization results of the original detailed data

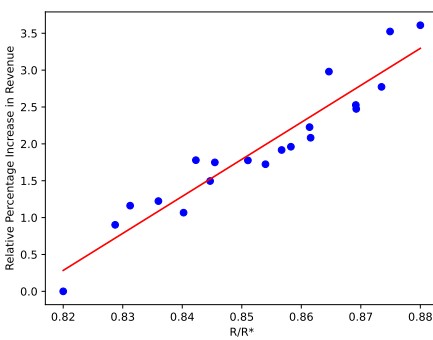

(a) The relationship between $R/R^*$ and the relative increase in online revenue. $R^2 = 0.902$, Avg. Dev. = 0.243, Max. Dev. = 0.483.

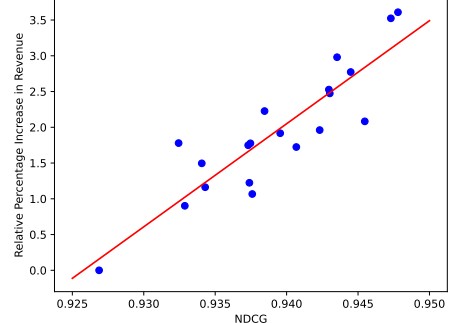

(b) The relationship between $NDCG$ and the relative increase in online revenue. $R^2 = 0.793$, Avg. Dev. = 0.311, Max. Dev. = 0.819.

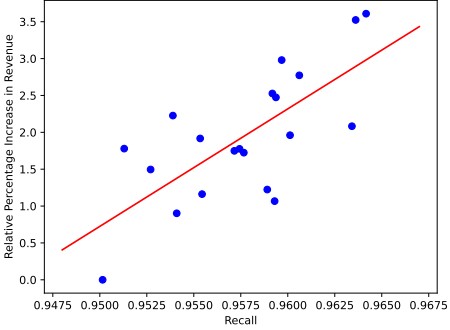

(c) The relationship between $Recall$ and the relative increase in online revenue. $R^2 = 0.509$, Avg. Dev. = 0.534, Max. Dev. = 1.139.

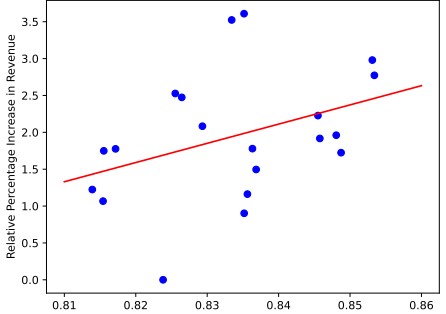

(d) The relationship between $OPA$ and the relative increase in online revenue. $R^2 = 0.140$, Avg. Dev. = 0.634, Max. Dev. = 1.690.

Figure 4: Relationships between evaluation metrics and online revenue **in the Pre-ranking stage** of our system. Each subplot shows the correlation with $R^2$, average deviation, and maximum deviation.

of online revenue and offline metrics. Figure 4 and Figure 5 shows the visualization results for the Pre-ranking and Matching stages of our system, respectively.

To protect commercial secrets, we applied a linear transformation to the original data, and the y-axis represents the relative increase in revenue rather than the absolute value. These operations do not affect the conclusions regarding the linear relationship. Each point in these figures represents data from one day of a model being live. **Note that all pairwise differences between points are statistically significant ($p < 0.05$), which is tested by the online A/B platform**.

We also observe that, in terms of the correlation between offline metrics and online revenue, the pre-ranking stage generally exhibits higher correlation coefficients compared to the matching stage. This may be attributed to the fact that, although both stages adopt a multi-pathway architecture, the matching stage involves a larger number of pathways. As a result, no single pathway dominates as strongly as in the pre-ranking stage. In the multi-pathway architecture, improvements in any individual pathway are more likely to be "covered" or offset by other pathways. This observation suggests that, in order to maintain a strong correlation between offline metrics and online performance, it is crucial to preserve dominant pathways with dominant weight within multi-pathway architectures.

### B.4 HYPER-PARAMETER SENSITIVITY ANALYSIS OF $R/R^*$

In the main text, we set $m = 2$ for $R/R^*$ based on a physical property of our system, rather than through validation-based hyperparameter tuning. Specifically, this value reflects the typical number of ads sent to the re-ranking stage for final selection in our training data pipeline. To further

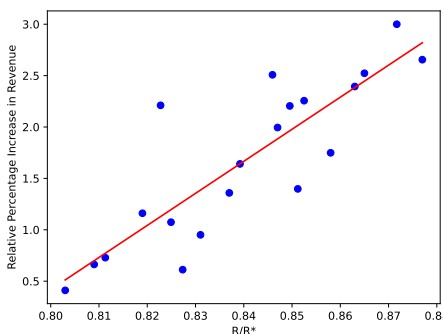

(a) The relationship between $R/R^*$ and the relative increase in online revenue. $R^2 = 0.725$, Avg. Dev. = 0.287, Max. Dev. = 1.082.

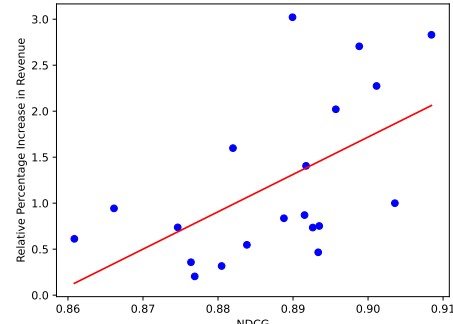

(b) The relationship between $NDCG$ and the relative increase in online revenue. $R^2 = 0.315$, Avg. Dev. = 0.626, Max. Dev. = 1.712.

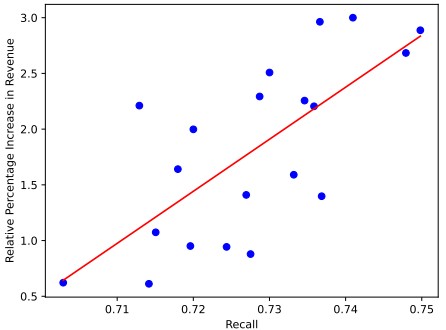

(c) The relationship between $Recall$ and the relative increase in online revenue. $R^2 = 0.504$, Avg. Dev. = 0.452, Max. Dev. = 1.100.

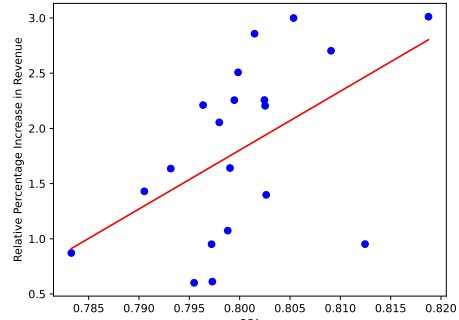

(d) The relationship between $OPA$ and the relative increase in online revenue. $R^2 = 0.259$, Avg. Dev. = 0.558, Max. Dev. = 1.515.

Figure 5: Relationships between evaluation metrics and online revenue **in the Matching stage** of our system. Each subplot shows the correlation with $R^2$, average deviation, and maximum deviation.

understand the impact of this choice, we conduct a sensitivity analysis of $R/R^*$'s performance under different values of the hyperparameter $m$.

In offline evaluations, only a limited number of ads are dumped for analysis, with 10 ads sampled per page view (PV, namely impression). Among these, 6 slots are reserved for fine-ranked ads, which are the only ones for which eCPM information is available. Consequently, we can only assess $R/R^*@m$ for $m$ ranging from 1 to 6. **For the pre-ranking stage**, the linear correlation coefficients ($R^2$) between these offline metrics and online revenue are reported as follows: $R/R^*(1) = 0.891$, $R/R^*(2) = 0.902$, $R/R^*(3) = 0.900$, $R/R^*(4) = 0.892$, $R/R^*(5) = 0.887$, and $R/R^*(6) = 0.830$. **For the Matching stage**, the corresponding linear correlation coefficients ($R^2$) are as follows: $R/R^*(1) = 0.666$, $R/R^*(2) = 0.725$, $R/R^*(3) = 0.699$, $R/R^*(4) = 0.695$, $R/R^*(5) = 0.698$, and $R/R^*(6) = 0.706$.

These results indicate that there indeed exists a reasonable $m$ that achieves a high linear correlation with online revenue. Specifically, among the evaluated metrics, $R/R^*$ demonstrates the highest correlation when $m = 2$, with a correlation coefficient of $0.9020$ and $0.725$ for the Pre-ranking and Matching stages, respectively. Moreover, it is noteworthy that the $R/R^*$ metric incorporates ad revenue information (eCPM) into the calculation of offline metrics, a feature not present in traditional metrics such as NDCG (Normalized Discounted Cumulative Gain) or Recall. Traditional metrics primarily focus on ranking accuracy and recall rates without directly accounting for the actual revenue generated by ads. Consequently, $R/R^*$ exhibits a stronger correlation with revenue across various values of $m$ compared to traditional metrics. We believe that the design of the $R/R^*$ metric holds significant reference value for the development of offline metrics in cascade ranking systems, especially those aimed at sorting based on a single value such as advertising revenue.

## C   MORE DETAILS AND DISCUSSIONS OF SCALING LAWS BETWEEN FLOPs AND $R/R^*$

### C.1   EXPERIMENTAL SETUP

#### C.1.1   TRAINING SETUP FOR MLPs AND DSSMs

**Regarding the features:** We adopt both sparse and dense features to describe the information of users and ads in the online advertising system. Sparse features are those whose embeddings are obtained from embedding lookup tables, while dense features are the raw values themselves. The sparse features of the user primarily include the action list of ads and user profile information (e.g., age, gender, and region). The action list mainly consists of action types, frequencies, target ads, and timestamps. The sparse features of the ads primarily include the IDs of the ad and its advertiser. The user's dense features mainly consist of embeddings produced by other pre-trained models. The dense features of the ads primarily include statistical features information, and multimodal features generated by multimodal models.

**Regarding the MLP models:** The MLP consists of 5 layers, where each hidden layer is composed of batch normalization, linear mapping, and a PReLU (He et al., 2015) activation function in sequence. The output layer is a pure linear layer. Parameters are initialized using the He initialization (He et al., 2015). We apply a log1p transformation as in (Qin et al., 2021) for all statistical dense features. All sparse and dense features are concatenated together and fed as input to the model.

**Regarding the DSSM models:** The DSSM consists of a user tower and an item tower. Each tower employs an MLP architecture and has 4 layers. Each hidden layer is composed of batch normalization, linear mapping, and a PReLU activation function. The output layer of the user tower employs only a linear mapping. The output layer of the item tower employs a linear mapping and an L2-normalization. The initialization strategy and feature processing methods are the same as those described for the MLP model.

Our model is trained online using a distributed training framework with synchronous training. We use Adagrad (Duchi et al., 2011) as the optimizer with a learning rate of 0.01 and an epsilon value of 1e-8. The training dataset comprises approximately 200 billion samples (i.e., user-ad pairs) and 20 billion impressions. The largest model, shown in Figure 3, was trained over 10 days using 60 A10 GPUs. The training tasks consume a data stream named $D_{\text{stream-train}}$, processing samples in the order they are generated. To ensure that all ads within the same impression are processed together for learning-to-rank training, the system groups ads by impression and performs shuffling at the impression level within small time windows.

#### C.1.2   TRAINING SETUP FOR TRANSFORMERS

To conduct transformer-based experiments, we follow recent practices in generative recommendation (Rajput et al., 2023; Zhai et al., 2024; Han et al., 2025; Deng et al., 2025; Zhou et al., 2025a;b) and adapt them to our advertising platform. We first extract multimodal embeddings using Qwen-VL 2.5 (Bai et al., 2025). To better align these multimodal embeddings with user behavioral signals in recommendation, we fine-tune and transform the embeddings following the QARM framework (Luo et al., 2024). Subsequently, we generate semantic IDs for all items using Res-kmeans.

We adopt a decoder-only architecture based on the open-source implementation of Qwen3 (Yang et al., 2025). Following Deng et al. (2025); Zhou et al. (2025a;b), we treat all impression logs as positive samples and employ full softmax loss for pre-training. The pre-trained model is then post-trained using a reward model derived from the online ranking system. Specifically, we use the production-grade ad ranking model as the reward model and apply an off-policy reinforcement learning strategy: the post-training data (including rewards) are collected from the live system, rather than generated by the model itself via inference. Since the reward model produces list-wise feedback, we employ the ARF (Wang et al., 2024) loss for post-training.

On the input side, item features consist solely of semantic IDs. For users, the input sequence includes behavioral sequences—such as clicks and conversions—represented by semantic IDs, along with user profile features (e.g., age, gender, context). The user profile features are placed at the beginning of the input sequence as a "prompt" to condition the model's predictions.

To ensure full model convergence, all Transformer models are trained on live production data for 25 consecutive days, and $R/R^*$ is evaluated on the subsequent day's data. For pre-training, we perform incremental batch training using daily-updated datasets, which facilitates efficient data compression similar to the approach in HSTU (Zhai et al., 2024). We denote this pre-training dataset as $D_{\text{pre-train}}$. For post-training, we initialize from the latest pre-trained checkpoint and perform streaming training using the same data stream (i.e., $D_{\text{stream-train}}$) as described in Appendix C.1.1.

Due to the significantly larger model size of our largest Transformer compared to MLP and DSSM models, we apply downsampling on $D_{\text{pre-train}}$ and $D_{\text{post-train}}$ to ensure that the daily training data of each scene can be fully processed within 24 hours using 64 flagship GPUs. We use the AdamW optimizer with a learning rate of 0.0001, weight decay of 0.01, $\beta_1 = 0.9$, $\beta_2 = 0.999$, and $\epsilon = 1\text{e}{-}8$. For evaluation, $R/R^*$ is computed over $D_{\text{stream-train}}$, where the predicted eCPM for each ad is obtained directly from the reward model's output score.

## C.2 More Experimental Results of Scaling Laws on Different Model Architectures

In addition to the key results presented in the main text, we provide extended experimental validation of scaling laws across multiple model architectures and system stages. We examine MLP and DSSM models across two business scenarios, confirming consistent scaling behavior between FLOPs and $R/R^*$. We further validate the existence of such scaling laws for Transformer-based architectures in the same industrial advertisement retrieval settings.

Here, we present additional results for the pre-ranking stage in $Scene_2$ using the MLP architecture, as shown in Figure 6. We also include results for the matching stage in $Scene_1$ and $Scene_2$ using the DSSM model, corresponding to Figure 7 and Figure 8, respectively. For Transformer-based models, we show the scaling curves on the matching stage in $Scene_1$ and $Scene_2$ in Figure 9 and Figure 10, respectively.

These results further support the existence of a broken neural scaling law in real-world industrial retrieval systems across diverse architectures. Importantly, this does not imply that the BNSL parameters are identical across architectures; rather, it means that within each architectural family, performance scales predictably with FLOPs in the observed FLOPs range.

## C.3 Discussion: Comparison with Scaling Laws in NLP

The scaling behaviors observed in our industrial advertisement retrieval system share formal similarities with those in NLP: both exhibit power-law relationships between model size (or FLOPs) and performance within certain computational ranges, enabling predictable performance trends. However, the practical application, scope, and implications of scaling laws differ significantly due to the distinct operational constraints of advertising systems.

A key difference lies in the **direction of extrapolation**. In NLP, scaling laws are typically used to *extrapolate upward*—predicting large-scale performance from small-scale experiments. In contrast, in advertising retrieval, it is often feasible to run large models on small traffic slices and then *infer their behavior at smaller, deployable scales*. This "downward inference" is enabled by the ability to safely test high-cost models offline or on limited traffic, making scaling laws a practical tool for ROI-driven model selection.

Another critical distinction is the **range of computation explored**. As exemplified by our experiments on the MLP architecture, the maximum FLOPs tested ($1.43 \times 10^{20}$) is approximately 20 times that of our online base model. While this range is far narrower than those in large-scale NLP studies (Kaplan et al., 2020; Hoffmann et al., 2022), it is sufficient for industrial deployment. This limitation arises not from methodological constraints, but from the economic reality of ROI optimization: beyond a certain point, the diminishing marginal returns implied by power-law scaling make further increases in model size unjustifiable. Even if performance continues to improve, the cost-benefit ratio becomes unfavorable.

Therefore, in the advertising domain, the value of scaling laws lies not in enabling ever-larger models, but in guiding efficient resource allocation and model design within practical budgets. We argue that future breakthroughs will likely stem not from scaling within existing paradigms, but from ar-

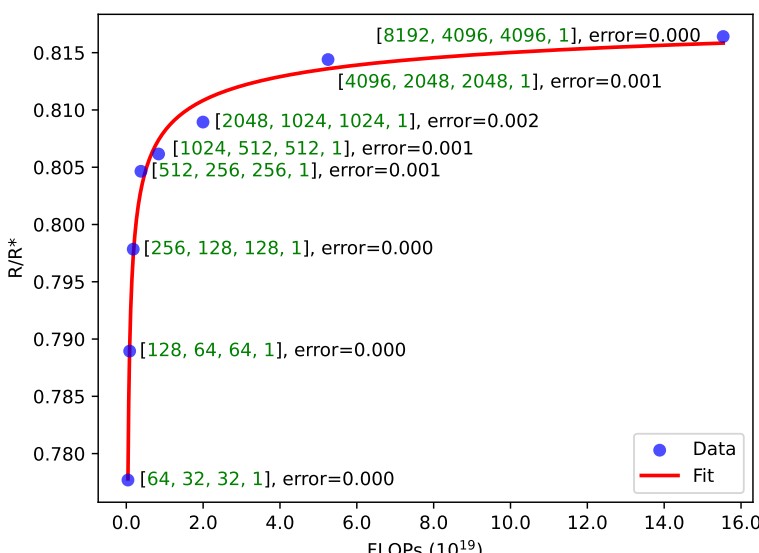

Figure 6: Scaling Performance regarding the FLOPs and $R/R^*$ of MLP models under $Scene_2$. The size of the MLP network and the deviation between the fitted curve and the true values (referred to as the "error") are annotated in the figure. The $R^2$ value of the curve fitting is 0.992. [1024, 512, 512, 1] indicates an MLP model with 4 layers, where the output sizes are 1024, 512, 512, and 1, respectively. The total dimensions of input features for all models are 3328.

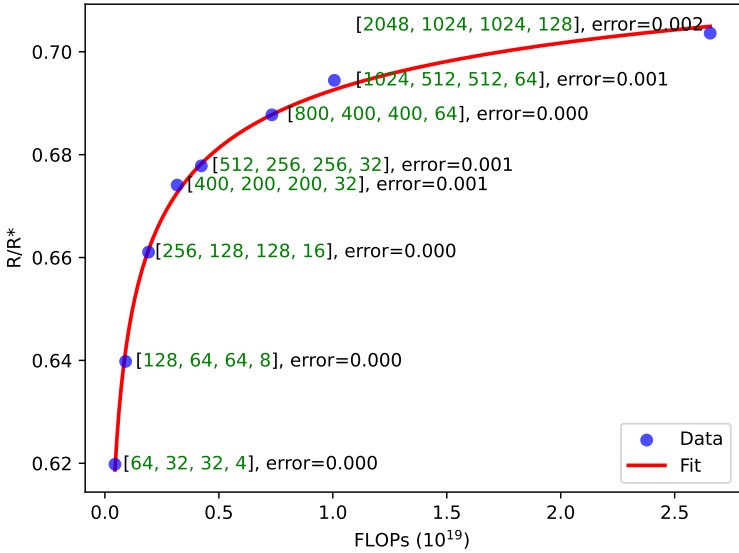

Figure 7: Scaling Performance regarding the FLOPs and $R/R^*$ of DSSM models under $Scene_1$. The size of the tower in DSSM and the deviation between the fitted curve and the true values (referred to as the "error") are annotated in the figure. The $R^2$ value of the curve fitting is 0.998. [1024, 512, 512, 64] indicates each tower of DSSM with 4 layers, where the output sizes are 1024, 512, 512, and 64, respectively. The total dimensions of input features for the user and item towers are 1760 and 1568, respectively.

chitectural or foundational innovations—what we term *paradigm innovations*—that shift the scaling curve itself. This perspective is further discussed in Appendix F.

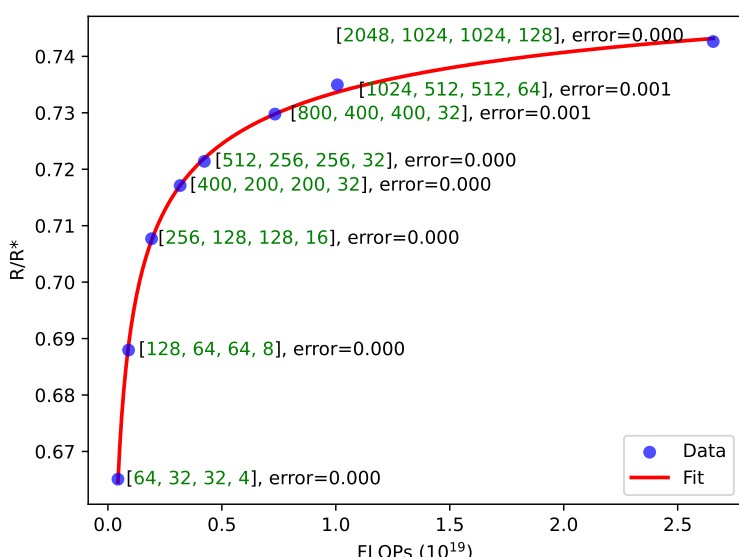

Figure 8: Scaling Performance regarding the FLOPs and $R/R^*$ of DSSM models under $Scene_2$. The size of the tower in DSSM and the deviation between the fitted curve and the true values (referred to as the "error") are annotated in the figure. The $R^2$ value of the curve fitting is 0.999. [1024, 512, 512, 64] indicates each tower of DSSM with 4 layers, where the output sizes are 1024, 512, 512, and 64, respectively. The total dimensions of input features for the user and item towers are 1760 and 1568, respectively.

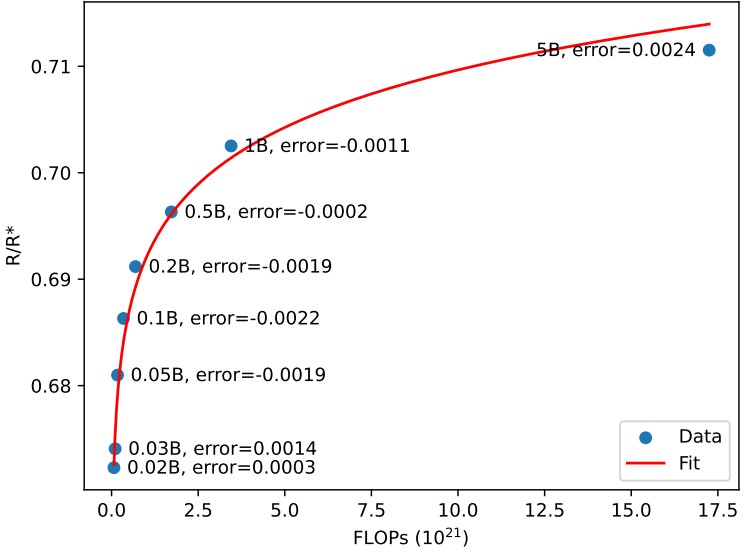

Figure 9: Scaling Performance regarding the FLOPs and $R/R^*$ of Transformer models under $Scene_1$. The size of Transformer models and the deviation between the fitted curve and the true values (referred to as the "error") are annotated in the figure. The $R^2$ value of the curve fitting is 0.984.

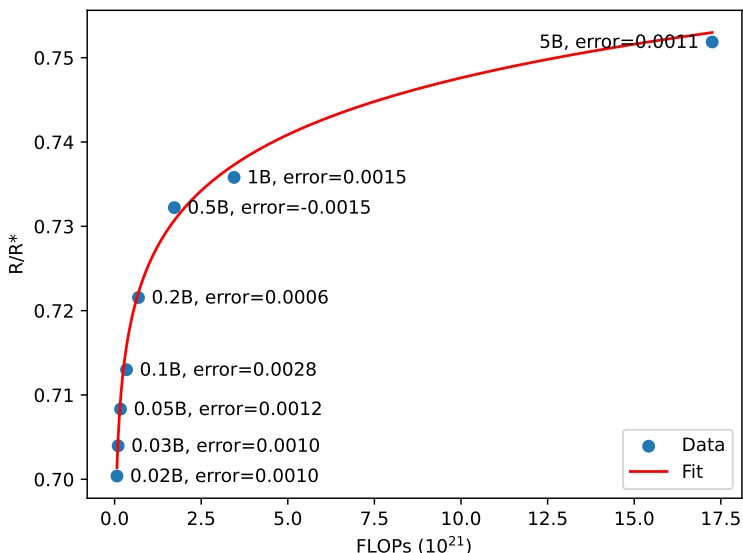

Figure 10: Scaling Performance regarding the FLOPs and $R/R^*$ of Transformer models under $Scene_2$. The size of Transformer models and the deviation between the fitted curve and the true values (referred to as the "error") are annotated in the figure. The $R^2$ value of the curve fitting is 0.992.

# D   MORE DETAILS ABOUT THE APPLICATIONS OF SCALING LAW

## D.1   DETAILS ABOUT INFRA SETTINGS FOR ONLINE TRAINING AND SERVING

Figure 11 illustrates the pipeline of online training and online serving in our advertising system and the details of "first-layer optimization" mentioned in section 3.3. The advertising system records logs in real-time for each ad exposure, including the features and labels of the training samples. The downstream training engine reads training data from Kafka storage in real-time and performs online training to optimize the model. Every 20 minutes, the training engine exports the latest model parameters to HDFS for secure storage. Meanwhile, the prediction server periodically polls HDFS to check for new model parameters and updates the local model parameters when a new version is available.

First-layer optimization is an optimization technique designed for MLP models. This technique transforms the computation of the first layer from $FFN(\text{concat}(emb_{user}, emb_{ad}))$ (referred to as $FL_{\text{naive}}$) to $FFN(emb_{user}) + FFN(emb_{ad})$ (referred to as $FL_{\text{opt}}$). The transformation yields equivalent results but significantly reduces computational overhead. This optimization is applied to the pre-ranking models of our advertising system, where each impression involves evaluating 1500 candidate ads. In this context, for each impression, there is one user embedding ($emb_{user}$) and 1500 ad embeddings ($emb_{ad}$). The operations concat and $+$ are broadcast operations, meaning they can be applied element-wise across arrays of different shapes. Here, the term "concat" refers to a logical operation with broadcasting properties, which in practice can be implemented using a combination of tf.tile and tf.concat in TensorFlow. By adopting the optimized form $FL_{\text{opt}}$, we can save a substantial amount of computation. To further enhance efficiency, we have developed a scheduled precomputation service called the "Embedding Producer." This service caches the $FFN(emb_{ad})$ values for all valid ads. It is a CPU-based service that is triggered by model update events and the insertion of new ads. This caching mechanism ensures that the precomputed values are readily available, reducing the computational overhead of the pre-ranking models. However, it also introduces a more complex relationship between the model's FLOPs and the associated machine costs. We use a TensorRT-based service architecture to deploy the model for online serving. To accelerate the computation during inference, we adopt half-precision floating point format (FP16) for each layer.

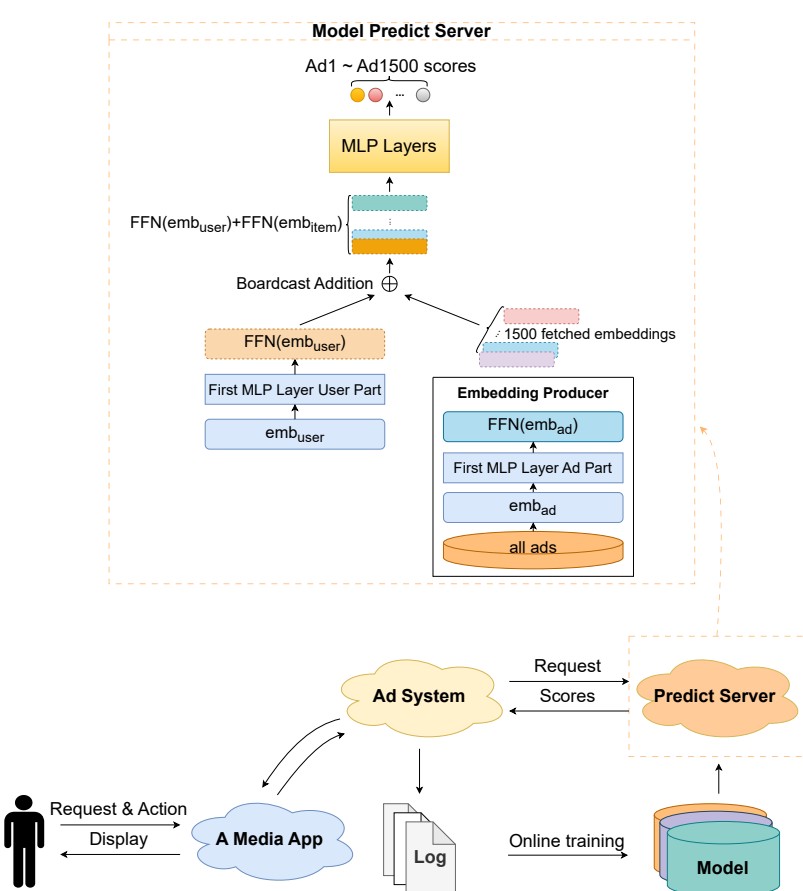

Figure 11: The pipeline of online training and serving, and details of the first layer optimization for Pre-ranking models.

For the training procedure of Pre-ranking models, we have also implemented the first-layer optimization. We use single precision floating point format (FP32) for training and saving the MLP and DSSM models. We use FP16 for training Transformer models. For training and serving DSSM and Transformer models in Matching stages, we incorporate several standard engineering optimizations to improve efficiency. These include model quantization, user tower computation compression, and caching of precomputed item tower embeddings. These techniques significantly reduce both the latency and the overall computational overhead during online training and serving. The inner product of the user and ad tower is directly calculated by brute force for online serving, and no approximate search algorithm such as FAISS is employed.

For both training and serving, we use T4 or A10 GPUs, with each GPU machine equipped with 128 CPU cores. Both the training and serving frameworks are developed based on TensorFlow (Abadi et al., 2016) or PyTorch (Paszke et al., 2019). Notably, the GPU models used for the experiments in Appendix C.1.2 differ from those used in standard online deployment. In Appendix C.1.2, to evaluate the performance of larger Transformer models, we utilize state-of-the-art, flagship-class GPUs rather than the T4 or A10 GPUs typically used in production. Due to company policies, certain detailed hardware specifications remain confidential and are not disclosed.

## D.2 AN DETAILED ILLUSTRATION OF MCET

In this section, we provide a detailed illustration of the Machine Cost Estimation Tool (MCET), which is implemented based on Algorithm 1. The MCET is designed to estimate the computation cost of deploying machine learning models without the need for actual online deployment. This tool

---

**Algorithm 1** Cost Estimation by Offline Simulation Testing

---

**Input:** The number $n$ of models (denoted as $n$), and meta files of the models (denoted as $\{meta_i | 0 \le i < n\}$). One specified machine $\mathcal{MH}$ for online serving. The machine number $req_0$ of the online base model $M_0$. The desired response time limit $T_{\text{limit}}$.

**Output:** The estimated machine number of each model for online serving, denoted as $\{req_i | 1 \le i < n\}$.

1: Initialize an empty list $results$ to store the results.
2: Simulate the execution of the base model $M_0$ using the specified hardware configuration $\mathcal{MH}$.
3: Measure the QPS ($QPS_0$) of the base model $M_0$ under the given response time limit $T_{\text{limit}}$, with the input being randomly initialized, fixed-length tensors.
4: Calculate the total QPS capacity of the base model setup: $QPS_{\text{total}} = req_0 \times QPS_0$.
5: **for** $i = 1$ to $n - 1$ **do**
6:    Load the meta file $meta_i$ of the $i$-th model.
7:    Simulate the model's execution using the specified hardware configuration $\mathcal{MH}$.
8:    Measure the QPS ($QPS_i$) of the $i$-th model under the given response time limit $T_{\text{limit}}$, with the input being randomly initialized, fixed-length tensors.
9:    Calculate the required number of machines for the $i$-th model: $req_i = \frac{QPS_{\text{total}}}{QPS_i}$.
10:    Append the result $req_i$ to the $results$ list.
11: **end for**
12: **return** $results$

---

is particularly useful in scenarios where rapid iteration and cost-effective deployment are critical, such as in large-scale advertising systems.

The core idea of MCET is to simulate the execution of models on a specified hardware configuration and measure their performance in terms of Queries Per Second (QPS) under a given response time limit. By leveraging these simulations, MCET can estimate the number of machines required to serve each model while meeting the desired performance constraints. This approach eliminates the need for costly and time-consuming online deployment, providing a reliable and efficient alternative for cost estimation. The algorithm begins by initializing a base model $M_0$ and simulating its execution on the specified hardware configuration $\mathcal{MH}$. The QPS of the base model $QPS_0$ is measured under the response time limit $T_{\text{limit}}$. Using this measurement, the total QPS capacity of the base model setup is calculated as $QPS_{\text{total}} = req_0 \times QPS_0$, where $req_0$ is the number of machines allocated to the base model.

Subsequently, the algorithm iterates over the remaining models, loading their meta files and simulating their execution on the same hardware configuration. For each model, the QPS $QPS_i$ is measured under the same response time limit, and the required number of machines $req_i$ is calculated as $req_i = \frac{QPS_{\text{total}}}{QPS_i}$. The results are stored in a list and returned as the final output.

The MCET tool, which implements this algorithm, allows us to estimate the machine requirements for tens of models within a short time frame. For instance, using a single machine with a single GPU, we can estimate the required number of machines for a single TensorFlow meta-file within half an hour. This capability significantly reduces the time and cost associated with model deployment and iteration.

Furthermore, by combining the estimated machine requirements with the unit price of the specified hardware configuration, we can accurately calculate the expected machine cost. This approach is applicable to both training and serving cost estimation, with the algorithm being executed using the meta files specific to each process. For training cost estimation, labels are randomly constructed to simulate the training environment.

In summary, MCET provides a standardized, reproducible, and efficient method for estimating computation costs, enabling rapid and cost-effective model deployment in real-world applications.

### D.3 ROI-CONSTRAINED MODEL DESIGNING

**Regrading Case Study 1:** We set an upper limit on the model FLOPs to 143M and set the step size of the grid search as 16 and 128 for the feature embedding dimensions and the unit number of

MLP layers, respectively. This larger step size is chosen because smaller increments have minimal impact on the total FLOPs and result in negligible increases in expected revenue according to the scaling law. The feature embedding dimension ($emb\_dim$) in the grid search starts at 16. The input size of MLP models is determined by multiplying the $emb\_dim$ by the number of sparse features, which is 200 in our experiments, plus an additional 128-dimensional dense feature. For the MLP layers (excluding the output layer, which is fixed at 1), the number of units starts at 128, with the constraint that the number of units in any layer does not exceed the number of units in the previous layer. The size of the first layer outputs does not exceed 1024, because this is the storage limit of the embedding producer. Additionally, we enforce that the number of units in any layer (except the output layer) is at least 1/20 of the number of units in the previous layer, as the reason discussed in appendix F.1. We used 10 T4-GPU machines to perform a grid search over approximately 1000 configurations and found an approximate optimal solution in about two days.

**Regrading Case Study 2:** We observe from the fitted data points that models in the 0.05B to 0.1B parameter range offer favorable cost-performance trade-offs. Leveraging the scaling law for Transformer-based models, we therefore conduct a grid search within this interval with a step size of 0.01B. Ultimately, a 0.08B-parameter model was deployed. Given that the model size is moderate, we are able to use A10 GPUs for both training and serving. Notably, the scaling law we fit remains in terms of FLOPs and $R/R^*$. When the average number of tokens per query is known, there exists a straightforward relationship between model parameter count (in billions) and FLOPs.

Additionally, we would like to give an additional clarification of the background in the application of ROI-constrained model designing: In the context of ROI-constrained model design, the application is premised upon a deliberate increase in machine cost. This is viable in scenarios where the baseline ROI provides sufficient margin above the business threshold $\lambda^*$, thereby sanctioning greater capital expenditure for incremental revenue. The value of $\lambda^*$ is set to ensure profitability, guaranteeing that the absolute revenue gain offsets the cost growth. Therefore, in both Case Study 1 and Case Study 2 discussed in Section 4.1 of the main text, the deployed models achieve improved performance by increasing machine cost relative to the base model, to obtain a net positive return on investment.

### D.4 MULTI-SCENARIO RESOURCE ALLOCATION

**Regrading Case Study 3:** As mentioned, we adopted an MLP model for the Pre-ranking stage and a DSSM model for the Matching stage. Through offline experiments, we obtained the key functions (including $G$, $BNSL$, etc.) in the scaling laws for both the Matching and Pre-ranking stages using our proposed paradigm. Due to the different model architectures, their machine costs vary in the same size. We used the machine cost estimation tool to additionally measure the machine costs of the DSSM model at different sizes. Similarly to the estimation of the MLP model, we solved the estimated machine costs of approximately 1000 different sizes of DSSM models based on the algorithm 1 using 10 T4-GPU machines for about 2 days.

**Regrading Case Study 4:** For $Scene_1$ and $Scene_2$, we perform proportional scaling of the base models in both the Pre-ranking and Matching stages. The scaling ratio ranges from 0.1 to 3.0 with a step size of 0.1. Based on the grid search results and the overall ROI across stages, we ultimately decide to reduce the model size in $Scene_1$ by 10% and increase the model size in $Scene_2$ by 50%.

## E  DISCUSSION OF THE INFLUENCE OF PREDICTION ERROR OF SCALING LAWS ON THE APPLICATIONS

A valid concern is whether the estimation errors in our framework—particularly in machine cost—could undermine the decisions made in our applications, especially when revenue relative gains are modest (e.g., +0.85% in Case Study 1). This appendix provides a formal analysis to demonstrate that our framework remains robust under the stated error bounds.

### E.1  ERROR PROPAGATION IN ROI ESTIMATION

The core decision metric in our applications is the Return on Investment (ROI), defined as:

$$\text{ROI} = \frac{R}{C},$$

where $R$ is the online revenue and $C$ is the machine cost.

Assume both the revenue prediction $\hat{R}$ and cost prediction $\hat{C}$ have a maximum estimation error of $\varepsilon$. The *actual* ROI thus lies within the interval:

$$\left[\frac{\hat{R}(1-\varepsilon)}{\hat{C}(1+\varepsilon)}, \ \frac{\hat{R}(1+\varepsilon)}{\hat{C}(1-\varepsilon)}\right] = \left[\widehat{ROI} \cdot \frac{1-\varepsilon}{1+\varepsilon}, \ \widehat{ROI} \cdot \frac{1+\varepsilon}{1-\varepsilon}\right].$$

For $\varepsilon = 5\%$, this simplifies to:

$$\text{Actual ROI} \in \left[\widehat{ROI} \times 0.905, \ \widehat{ROI} \times 1.105\right],$$

indicating an **uncertainty window of approximately -9.5% to +10.5%** around the estimated ROI.

### E.2    ROBUST DECISION-MAKING UNDER UNCERTAINTY

A candidate model can be confidently selected if it satisfies two conditions under this uncertainty:

1. The **upper bound** of its estimated ROI remains **below** the baseline model's ROI.
2. The **lower bound** of its estimated ROI remains **above** the business-mandated threshold $\lambda^*$.

### E.3    REPRESENTATIVE NUMERICAL EXAMPLE

Consider a scenario reflective of our ROI-constrained model design application (Case Study 1):

- **Baseline Model (Deployed):**
    - Actual Revenue, $R_{\text{base}} = 100.00$
    - Actual Cost, $C_{\text{base}} = 10.00$
    - Actual ROI, $\text{ROI}_{\text{base}} = 10.00$
- **Candidate Model (Proposed):**
    - Actual Revenue, $R_{\text{opt}} = 100.85$ (**+0.85% gain**)
    - Actual Cost, $C_{\text{opt}} = 15.00$
    - Actual ROI, $\text{ROI}_{\text{opt}} = 100.85/15.00 \approx 6.72$

Given the 5% estimation error, the **estimated ROI range** for the candidate model is:

$$[6.72 \times 0.905, \ 6.72 \times 1.105] \approx [6.08, \ 7.43].$$

In a conservative (worst-case) scenario where the *estimated* ROI is at the top of this range (7.43), the corresponding *actual* ROI range would be inferred as in the following range:

$$[7.43 \times 0.905, \ 7.43 \times 1.105] \approx [6.72, \ 8.21].$$

Assuming a business threshold $\lambda^* = 6$, we observe:

- The **lower bound** of the actual ROI (6.72) is still significantly greater than $\lambda^*$ (6).
- The model delivers a **net positive revenue gain** of 0.85%.

This demonstrates that even with a 5% cost error, the decision to deploy the candidate model is robust and value-adding.

### E.4    CONCLUSION

This analysis shows that a non-negligible estimation error does not preclude reliable decision-making. The success of our applications hinges on identifying configurations where the predicted improvement is significant enough to be robust within the error bounds, a condition satisfied in our system as evidenced by the online results. The error levels reported in our work represent a level of accuracy that is both practically achievable and sufficient for industrial optimization. Additionally, reducing these estimation errors would allow us to identify more precise and closer-to-optimal model designs.

# F LIMITATIONS AND FUTURE WORK

## F.1 OPTIMAL MODEL DESIGNING WITH GIVEN FLOPs

One remaining question is how to give optimal unit distributions of layers under a given FLOPs. Typically, in advertising models, the size of the bottom layer is the largest, gradually decreasing in the upper layers. We believe that as long as the difference in the number of units between adjacent layers is within a reasonable range, the performance impact should be negligible. To explore this, we conducted preliminary experiments, such as testing a model with the size [3328, 1024, 32, 32, 32, 1]. We found that when the unit allocation is unreasonable, the model performance significantly degrades compared to a reasonable one. Our empirical experience suggests that, except for the output layer, the difference in the number of units between adjacent layers should not exceed 20. Models designed following this principle generally conform to the BNSL. More in-depth conclusions are left for future work and are beyond the scope of this paper.

## F.2 JOW LAWS IDENTIFICATION AND APPLICATION

Obtaining joint laws considering multiple factors, such as data, model size, and computational resources, is significantly more costly in industrial advertising settings than obtaining single-variable scaling laws. Developing a cost-effective method for deriving joint laws is therefore an important area for future investigation.

An especially promising direction for application is to model the joint law between **model size** and the **computational quota allocation** across stages in a cascade ranking system. Such a law could help answer critical questions: How should we rebalance ranking, pre-ranking, and matching stage budgets when scaling up the overall model? At what point does increasing model size in one stage yield diminishing returns due to bottlenecks in another? By formalizing these interactions, joint laws can deepen our understanding of cascade systems and guide principled, globally optimal configuration decisions with only a given budget.

## F.3 BEYOND COMPUTE SCALING

Although our optimization framework has achieved an overall 5.1% improvement in ad revenue, a substantial portion of this gain arises not from breakthroughs in model architecture, but from correcting historically suboptimal resource allocation and coarse-grained manual tuning. Our results indicate that, unlike in NLP, simply scaling up computation alone is unlikely to bring similar levels of success to the advertising domain for models such as MLP, DSSM, or Transformer.

Therefore, the next major leap in performance will likely require more disruptive innovations. One path is architectural rethinking—designing models with inherently better scaling efficiency, such as sparse, modular architectures (Liu et al., 2024). Another is a data-centric paradigm shift: leveraging large language models or product-level innovations to improve the *quality*, *semantic richness*, and *diversity* of training signals. In this view, "scaling" must evolve beyond model size to include the scaling of data intelligence. Only through such foundational changes can we hope to achieve the kind of sustained, huge progress seen in NLP domains.

# G REPRODUCIBILITY STATEMENT

Our experiments are conducted on an industrial online advertising platform. Due to company policies, we are unable to release the raw data or certain proprietary code. However, we have made substantial efforts to ensure the reproducibility and transferability of our methodology and findings. Specifically:

- **The Lightweight Scaling Law Identification Framework:** In Section 3.1, we provide a detailed definition and computation procedure for the $R/R^*$ metric. Additionally, Algorithm 1 and Appendix D.2 fully describe the Machine Cost Estimation Tool (MECT), allowing researchers to adapt our framework to other systems or domains.

- **Empirical Validation of Scaling Laws:** To support the reproducibility of our scaling law observations for models such as MLP, DSSM, and Transformer in advertisement retrieval, we provide comprehensive experimental setups in Appendix C.

- **Applications of Scaling Laws:** For the practical applications discussed in the paper, we document the full setup in Section 4 and Appendix D. This includes infrastructure specifications for training and serving, formulation of the optimization problems, and solution procedures.

We hope these materials significantly lower the barrier for reproducing and extending our work in both academic and industrial settings.

## H    LLM USAGE STATEMENT

We used large language models (LLMs) as a general-purpose writing assistance tool during the preparation of this paper. Specifically, we leveraged LLMs to improve the clarity, grammar, and fluency of the manuscript. All technical content, including research ideas, methodology, analysis, and conclusions, was developed solely by the human authors. The LLM was not involved in research design, data interpretation, or scientific decision-making. We have reviewed and verified all text generated or modified with the help of the LLM to ensure accuracy and originality. The use of the LLM does not constitute authorship, and the responsibility for the integrity of the work rests entirely with the listed authors.

