# OpenReview forum: "Scaling Laws for Online Advertisement Retrieval"
_ICLR.cc/2026/Conference — Submitted to ICLR 2026_

### Official Review · Reviewer_ohso · 2025-10-27

**Soundness:** 2
**Presentation:** 2
**Contribution:** 2
**Rating:** 2
**Confidence:** 4

**Summary:**

The paper aims to establish laws based on empirical observations for predicting how would revenue change given a retrieval model's architectural parameters. The paper devises a metric called $R/R*$ that is claimed to correlate with revenue, propose fitting a BNSL-type scaling law to $f(\mathrm{FLOPs) \approx R/R*$, and then use the model's architecture to compute costs in order to estimate the cost-effectiveness of a model.

**Strengths:**

1. The problem is important. I am also not aware for good scaling laws for ad retrieval models, where the final ranking is **not** done by the model's predictions, but via the expected revenue.
2. The initial presentation of an ad ranking system is good and puts the paper into context.
3. Benchmarks do show some predictive power of the proposed laws.

**Weaknesses:**

1. The paper lacks soundness and convincing arguments. Here are a few examples:
   1. the training data talks about a vector $v$ of "ground-truth order", but how is this order determined? I guess it's by the eCPM, since I personally come from the ad industry and understand that the job of pre-ranking is selecting the candidates that have the highest probability of (I come from the advertising business myself), but it's not obvious what is the ground-truth order of a pre-ranking system.
   2. The FLOPs scaling laws may be convincing for language models, where we have established practices for many architectural free variables, such as the fact that MLP layers have intermediate layer of 4 x embedding dimension, and others. But for a generic MLP, I do not see FLOPs as a reasonable proxy for scaling, unless you add architectural assumptions as well. And who said these assumptions are reasonable in general for ad ranking models?
  3. How exactly can A/B tests establish linear correlation? In A/B tests you're testing two candidates against each other. Here you just need to test many models, not against each other. If you **are** testing them against each other, it's the rank correlation that matters. If you're just collecting revenue samples, these aren't A/B tests. An explanation is in place here.
2. The training data is biased. You know the ground-truth order only for those that actually were selected by pre-ranking, but you don't know the order of those that weren't. Or are you constructing the training data synthetically not from real ad auctions? An explanation is missing. Either how you account for the bias, or how is the training data constructed.

**Questions:**

See weaknesses.

---

> ### Author Response · Authors · 2025-11-16
>
> Dear Reviewer ohso,
>
> Thank you for your review and for your positive comments regarding the importance of our problem setting and the demonstrated predictive power of our scaling laws. We address your specific concerns below.
>
> **For weakness 1.1):**
>
> The ground-truth rank is introduced in Section 2.2. Specifically, the rank of ad ( i ) is higher than that of ad ( j ) if ad ( i ) comes from a later stage than ad ( j ), or if they are from the same stage and ad ( i ) is ranked higher within that stage.
>
> An important clarification: for ads sampled from the Ranking stage, their internal order is indeed determined by their predicted eCPM. However, for ads sampled from the Matching and Pre-ranking stages, the internal order is determined by the predicted scores from their respective stage models. While these scores lack the direct monetary interpretation of eCPM, a higher score is still treated as indicating better quality for ranking purposes within that stage.
>
> This formulation for constructing a global, cross-stage ground-truth order follows the sample construction of prior works [1–3]. We will expand on this explanation in the revised manuscript to improve clarity.
>
> **For weakness 1.2):**
>
> Thank you for raising this important concern. We understand your point that in NLP, Transformer models often scale in a proportional and highly structured manner, whereas for a generic MLP, the relationship between FLOPs and scaling behavior may appear less straightforward without additional architectural priors.
>
> Indeed, we acknowledge this issue and have discussed it in Appendix E.1 (Limitations and Future Work). In our work, we observe that MLP scaling laws remain consistent and predictable only when the model designs adhere to certain basic constraints—for example, in our empirical setting, the size ratio between consecutive layers should not exceed a factor of 20. Under such constraints, FLOPs can serve as a reliable scaling variable across different MLP configurations.
>
> In practical advertising applications, when scaling a base model, we typically follow a near-proportional scaling strategy. Therefore, we believe the concern regarding the absence of FLOPs-based scaling laws is mitigated in real-world usage.
>
> That said, we wish to emphasize that the core contribution of our work lies in proposing a lightweight paradigm for identifying scaling laws in industrial environments and demonstrating their potential to guide model design under real system constraints. We have provided a thorough discussion of the limitations and future directions in Appendix E.
>
> **For weakness 2):**
>
> In Appendix B.2 (Lines 830-834), we describe: “Due to the limitations on experimental traffic, we can only run up to 5 A/B test groups simultaneously, which means some experimental groups were launched on different days. The online experiments corresponding to each model use at least 5% of the traffic. We collect each model’s online revenue from the A/B test platform and offline metrics from the training log.”
>
> We calculate the relative improvement in online revenue for each model compared to this baseline. Since we always keep a baseline model online, this allows models launched on different days to remain comparable, ensuring that we accurately capture the relationship between online revenue and $R/R^\*$ despite limited traffic resources. When a strong linear relationship exists between the relative improvement in online revenue and $R/R^\*$, the same should logically hold for the absolute online revenue.
>
> We understand that you may consider the definition of an A/B test to strictly refer to a comparison between two groups: group A and group B. However, since these experiments rely on the online A/B testing platform, we refer to them as A/B test-based. We will carefully revise the manuscript to clarify this point.
>
> **For weakness 3):**
>
> Thank you for raising this important point regarding selection bias. Selection bias is a well-known characteristic of cascade ranking systems. We agree that this is an inherent challenge in any multi-stage recommender system, and any method applied to such systems must operate within this constraint.
>
> In our work, we adopt the full-stage sampling strategy to construct our training data, as detailed described in our response to Weakness 1.1. This approach, which leverages samples across all stages, has been shown to provide theoretical grounding [1] for alleviating the selection bias inherent in cascade systems.
>
> Thank you again for your time and effort in reviewing our work. If any concerns remain, we are fully committed to addressing them in our follow-up responses.
>
> **References**
>
> [1] Zheng et al. Full Stage Learning to Rank: A Unified Framework for Multi-Stage Systems. WWW, 2024.
>
> [2] Wang et al. Adaptive Neural Ranking Framework: Toward Maximized Business Goal for Cascade Ranking Systems. WWW, 2024.
>
> [3] Wang et al. Learning Cascade Ranking as One Network. ICML, 2025.

---

> > ### Comment · Reviewer_ohso · 2025-11-19
> > **Raising score**
> >
> > Thank you for your response.
> >
> > Most of my concerns were addressed, except for FLOPS being a reasonable proxy, which I believe has not been adequately addressed.  The claims about your architectural constraints being "typical" aren't convincing - I come myself from the ad industry, and it's quite a wild-west - there's nothing "typical". Nevertheless, it may be the case that there are other systems with MLP having the architectural constraints you chose, and as long as you explicitly clarify your way of scaling, and aknowledge that the scaling laws are a good fit for this specific way - I believe the contribution should be accepted.

---

> > > ### Author Response · Authors · 2025-11-20
> > >
> > > Thank you for pushing us to clarify this critical point. We accept your suggestion and will refine the manuscript to precisely scope our claims.
> > >
> > > Specifically, we will **add a clear statement in Section 3.2** that the identified scaling laws for MLPs are demonstrated **within a specific, constrained design space**: namely, **near-proportional scaling** where the size ratio between consecutive hidden layers is kept within a factor of 20. **We affirm that the scaling laws are a good fit specifically for this strategy**.
> > >
> > > We maintain that this constrained design space is highly relevant industrially, as it reflects the common practice of proportionally scaling successful base models (evidenced by the configurations in Figs. 3, 6-8). Within this well-defined and practical space, our work demonstrates that FLOPs serve as a powerful and reliable proxy.
> > >
> > > We hope these precise claims will resolve your concern and underscore the practical contribution of our paradigm.
> > >
> > > Thank you again for your prompt response and constructive feedback!

---

### Official Review · Reviewer_UjMm · 2025-10-31

**Soundness:** 3
**Presentation:** 1
**Contribution:** 2
**Rating:** 2
**Confidence:** 3

**Summary:**

This paper investigates whether scaling laws also apply to industrial online advertising systems. The authors propose a lightweight offline paradigm to identify scaling laws between machine cost and online revenue for ad retrieval models. They introduce a new offline metric, R/R*, to be linearly correlated with online revenue. Using FLOPs as the scaling factor, the study confirms broken neural scaling laws across Transformer, MLP, and DSSM architectures in real-world ad retrieval systems. They further develop a machine cost simulation tool and apply the framework to ROI-constrained model design and multi-scenario resource allocation.

**Strengths:**

As real-world advertising systems become increasingly complex and large-scale, it is important to make changes that comply with ROI constraints. This paper proposes a heuristic scaling law to study the trade-off between cost and return. Overall, the paper is interesting and useful.

**Weaknesses:**

My main concern is that the writing could be substantially improved, as the current version of the paper is not accessible to the general ICLR audience.

1. The key concept $R / R^*$ is mentioned several times in the Introduction and Abstract, but it is never explained, even heuristically.
2. The definition of $R / R^*$ in Equation (2) is difficult to understand. What is the “hard permutation matrix”? It should be clearly defined, and a heuristic explanation would be helpful.
3. How is the ground-truth rank $v_i^j$ obtained? Is it derived from the final ranking stage?
4. The statement “If the following assumptions are met, we can prove that $R / R^*$ and online revenue have a linear relationship” is never formally presented as a theorem.
5. Typos: “Assumption Theorems” on Line 256; “Section Sections” on Line 36.

Moreover, I have additional comments on the paper:
6. I do not think the linear relationship between $R / R^*$ and online revenue qualifies as a scaling law. In my view, Equation (4) represents a scaling law, but it is only heuristic.

7. In the case studies and applications, only the revenue gains are reported. However, given that the paper focuses on ROI, it would be important to also report the corresponding cost comparisons.

**Questions:**

See above

---

> ### Author Response · Authors · 2025-11-16
>
> Dear Reviewer UjMm,
>
> Thank you for your thoughtful review and for recognizing our work as interesting and useful. We appreciate your constructive feedback and have prepared detailed responses to each of the weaknesses you raised.
>
> **1) Explanation of ( $R/R^\*$ )**: We provided a heuristic explanation of ( $R/R^\*$ ) in Lines 75–76 and a detailed formal definition in Section 3.1. In the revised manuscript, we will further improve the intuitive description to make it more accessible to a broader audience.
>
> **2) Definition of “Hard Permutation Matrix”**: The concept of a hard permutation matrix is widely used in differentiable sorting literature. Due to space limitations, we referenced NeuralSort (Line 225), which introduces this concept. For example, given a vector $\mathbf{a} = [2, 4, 1, 3]$, the sorted version is $\mathbf{b}=[4, 3, 2, 1]$, and the hard permutation matrix can be constructed as $P = \begin{bmatrix}
> 0 & 1 & 0 & 0 \\\\
> 0 & 0 & 0 & 1 \\\\
> 1 & 0 & 0 & 0 \\\\
> 0 & 0 & 1 & 0
> \end{bmatrix}$ accordingly, where satisfies $\mathbf{b}=P\mathbf{a}$. We will include a concise illustrative example in the revision to clarify this concept.
>
> **3) Ground-Truth Rank Definition**: The ground-truth rank is introduced in Section 2.2. Specifically, the rank of $ad_i$ is higher than that of $ad_j$ if $ad_i$ comes from a later stage than $ad_j$, or if they are from the same stage and $ad_i$ is ranked higher within that stage. This formulation follows the sample construction of prior works [1–3]. To improve clarity, we will expand on this explanation in the revised manuscript.
>
> **4) Theoretical Presentation**: The assumptions are stated in Section 3.1, and the proof is provided in Appendix B.1. Following your suggestion, we will formally present this result as a theorem in the main text, clearly stating that under the given assumptions, ($R/R^\*$) and online revenue exhibit a strict linear relationship.
>
> **5) Typos and Errors**: Thank you for carefully identifying the typos (e.g., “Assumption Theorems” in Line 256, “Section Sections” in Line 36). We will correct these and perform a thorough proofreading of the entire manuscript to eliminate any further errors.
>
> **6) On the Nature of Scaling Laws**: We aim to model the scaling law between FLOPs and online revenue. The metric ($R/R^\*$) serves as an offline surrogate to reduce experimental cost. The relationship between FLOPs and ($R/R^\*$) follows a Broken Neural Scaling Law (BNSL) [4], and ($R/R^\*$) is approximately linearly correlated with online revenue. Together, these support the existence of a scaling law between FLOPs and online revenue, which also aligns with the BNSL form. We agree that scaling laws are inherently empirical and heuristic in nature, as also noted in [4-6].
>
> **7) Machine Cost and ROI Reporting in Applications**: All applications were conducted under strict ROI constraints and fixed total machine budgets. However, specific values of ROI thresholds and machine costs are considered sensitive business information and cannot be disclosed. We believe these numerical values are not critical to the methodological contribution, as different scenarios will have unique budgets and ROI constraints. Our framework is general and can be applied to diverse scenarios with varying constraints.
>
> Thank you again for your time and effort in reviewing our work. If any concerns remain, we are fully committed to addressing them in our follow-up responses.
>
> **References**
>
> [1] Zheng et al. Full Stage Learning to Rank: A Unified Framework for Multi-Stage Systems. WWW, 2024.
>
> [2] Wang et al. Adaptive Neural Ranking Framework: Toward Maximized Business Goal for Cascade Ranking Systems. WWW, 2024.
>
> [3] Wang et al. Learning Cascade Ranking as One Network. ICML, 2025.
>
> [4] Caballero et al. Broken Neural Scaling Laws. ICLR, 2023.
>
> [5] Kaplan et al. Scaling laws for neural language models. arXiv preprint arXiv:2001.08361, 2020.
>
> [6] Hoffmann et al. Training compute-optimal large language models. arXiv preprint arXiv:2203.15556, 2022.

---

> > ### Comment · Reviewer_UjMm · 2025-11-16
> > **Thank you for the authors' clear and prompt response.**
> >
> > First, thank you for the authors' clear and prompt response.
> > 1. Lines  75–76 read *First, we introduce a novel offline metric, R/R∗,which integrates the predicted revenue of each ad from the training data.* I do not think it is qualified as heuristic explanation.
> > 3.  Thank you for the further explanation regarding the ground-truth rank. However, I did not find the details of the ground-truth rank in Section 2.2.
> >
> > 7. Do you mean that for all metrics reported, the ROIs are the same?

---

> > > ### Author Response · Authors · 2025-11-16
> > >
> > > Dear reviewer,
> > >
> > > **1) Regarding the heuristic explanation of $R/R^\*$**
> > >
> > > We agree that while Section 3.1 provides a detailed formulation of $R/R^\*$, offering a more intuitive explanation in the introduction would help readers from broader backgrounds grasp its essence more quickly. We sincerely appreciate your suggestion. Considering space limitations, we plan to add a concise and clear description:
> > >
> > > " $R/R^\*$ is evaluated on samples collected from the full-stage pipeline. Its advantage over traditional metrics lies in its ability to better simulate the real online environment and its incorporation of eCPM information, which is highly correlated with revenue."
> > >
> > > (Note: The high correlation between eCPM and revenue is a well-established background knowledge in advertising systems.)
> > >
> > > Would this be sufficient, or do you have any specific suggestions for wording? We are very open to your guidance.
> > >
> > > **2) Regarding the details of the ground-truth rank in Section 2.2:**
> > >
> > > In Lines 187–189, we describe how the ground-truth rank is defined: *"The rank index is the relative position of the sampled ads within the system queue, ordered by their original positions."* We acknowledge that this may have been too brief to attract the reader's attention. In the next version, we will formalize the label definition mathematically, for example as:
> > >
> > > $\mathbb{1}(v_i > v_j) = \mathbb{1}(s_i > s_j) \lor \left( \mathbb{1}(s_i = s_j) \land \mathbb{1}(\text{score}_i > \text{score}_j) \right)$
> > >
> > > where $s$ denotes the sampling stage of the ad, and $\text{score}$ represents the system-assigned score within that stage.
> > >
> > > **3) Regarding whether ROIs are the same for all metrics:**
> > >
> > > We may not have fully understood the question. The ROI constraints are specific to the **applications** (i.e., in Sections 4.1 and 4.2). For example, in Line 426, the deployed model with size [16128,1024,512,512,512,1] just satisfies the ROI constraint of our scenario, while the ROI of baseline model with size [3328,1024,256,256,256,1] is much bigger than the ROI constraint of our scenario.
> > >
> > > However, for the figures of scaling curves (Figs. 3, 6–9), each point represents a model with a **different configuration**, and thus their ROIs are certainly not the same.
> > >
> > > Thank you very much for your prompt response! If this clarification still does not fully address your question, could you please elaborate further on what you meant?

---

> > > > ### Comment · Reviewer_UjMm · 2025-11-20
> > > > **Regarding ROIs**
> > > >
> > > > For example, in case study 1, do you mean  [16128, 1024, 512, 512, 512, 1] and the baseline [3328, 1024, 256, 256, 256, 1] have the same ROIs as $\lambda^*$?
> > > >
> > > > Furthermore, as the e G◦BNSL◦F and MCET are only approximated, the real cost comparison is also worth evaluating.

---

> > > > > ### Author Response · Authors · 2025-11-20
> > > > >
> > > > > Thank you for your follow-up questions. We are happy to provide further clarification on these key points.
> > > > >
> > > > > ### 1. Clarification on ROI in Case Study 1
> > > > >
> > > > > The relationship between the models' ROIs and the threshold `λ*` is fundamental to our application. Here is a precise breakdown:
> > > > >
> > > > > -   **`λ*` is a minimum constraint, not a target.** It is a business-defined threshold that any deployed model must meet or exceed.
> > > > > -   **ROI is defined as:** `Online Revenue / Machine Cost`.
> > > > > -   **Typical Scaling Behavior:** When scaling up a model, the increase in revenue often diminishes relative to the increase in cost. Therefore, it is common for a larger, more powerful model to have a **lower ROI** than a smaller, more efficient baseline.
> > > > > -   **Case Study 1 Scenario:**
> > > > >     -   Let `A` = ROI of baseline model `[3328, 1024, 256, 256, 256, 1]`
> > > > >     -   Let `B` = ROI of optimized model `[16128, 1024, 512, 512, 512, 1]`
> > > > >     -   The common scenario is: `A > B ≥ λ*`
> > > > > -   **Our Goal:** We did not seek a model with an ROI equal to `λ*`. We sought the model that **maximized online revenue** from the set of all models whose ROI satisfied `ROI ≥ λ*`. The optimized model was selected because it provided a **+0.85% revenue gain** while clearing the minimum ROI bar.
> > > > >
> > > > > ### 2. Validation of the Cost-Revenue Framework
> > > > >
> > > > > Our framework is designed for industrial applications, where predictive accuracy is critical.
> > > > >
> > > > > -   **Revenue Prediction Accuracy:** As empirically validated in Section 3.2, the chain from `FLOPs → R/R* → Revenue` is highly accurate (e.g., 0.33%, 0.62% predicted vs. 0.38%, 0.69% actual gain). This predictive power is further supported by the high goodness-of-fit at each stage: the FLOPs → R/R* relationship follows a BNSL with R² > 0.98, and the R/R* → Online Revenue correlation has an R² of 0.902 (Pre-ranking) and 0.725 (Matching).
> > > > > -   **Cost Prediction Accuracy:** Our Machine Cost Estimation Tool (MCET) is not a heuristic but a system-level simulator. In our production environment, its **estimated machine cost is consistently within 5% of the actual cost**, a level of accuracy that is fully sufficient for industrial resource planning and ROI calculations. We will add this performance metric to Section 3.3.
> > > > > -   **End-to-End Validation:** The success of our ROI-constrained model designing (Section 4.1) and multi-scenario resource allocation (Section 4.2), which achieved a **significant revenue gain under a fixed total machine budget or ROI constraint**, serves as the ultimate proof that our integrated framework provides reliable, actionable predictions for system optimization.
> > > > >
> > > > > We hope these clarifications fully address your questions.

---

> > > > > > ### Comment · Reviewer_UjMm · 2025-11-21
> > > > > > **Cost estimation**
> > > > > >
> > > > > > A 5% error in cost estimation is not a small figure, especially when you consider that the revenue gain in Case 1 is only 0.85%. The potential impact of such an error could be significant.

---

> ### Author Response · Authors · 2025-11-21
>
> Thanks very much for your prompt feedback! Your question is interesting; the numerical difference between 0.85% and 5% might indeed raise such concerns at first glance. However, a maximum error of 5% does not mean that a 0.85% increase in revenue is unattainable in industrial applications. We will illustrate this through mathematical analysis and a practical example.
>
> **I. Mathematical Analysis of Error Propagation**
>
> The core of our decision is the relative ROI between models. Let's define:
> - **R**: Online Revenue
> - **C**: Machine Cost
> - **ROI = R / C**
>
> Let's assume both our revenue and cost predictions have a maximum error of **ε = 5%**. The *actual ROI* therefore lies within the range:
> `[ R_est * 0.95 / (C_est * 1.05) , R_est * 1.05 / (C_est * 0.95) ] = [ Estimated_ROI * (0.95/1.05) , Estimated_ROI * (1.05/0.95) ]`
>
> This simplifies to: `[ Estimated_ROI * 0.905 , Estimated_ROI * 1.105 ]`
>
> This means the ROI estimate has an **uncertainty window of roughly -9.5% to +10.5%**.
>
> **II. Influence in Application and a Numerical Example**
>
> A candidate model can be confidently selected as superior if it satisfies two conditions simultaneously:
> 1.  The **upper bound** of its estimated ROI (factoring in maximum error) is **less than the baseline model's ROI**.
> 2.  The **lower bound** of its estimated ROI remains **greater than the business-mandated threshold λ***.
>
> We will now demonstrate with a representative example that such a scenario is possible and can occur in practice, even with a cost estimation error of 5% and a revenue gain of only 0.85%.
>
> **A Representative Numerical Example**
>
> Consider the following scenario, constructed to reflect our decision-making process:
>
> - **Baseline Model:**
>     - Actual Revenue: `R_base = 100.00`
>     - Actual Cost: `C_base = 10.00`
>     - Actual ROI: `ROI_base = 10.00`
>
> (Since the baseline model is already deployed in the production environment, its actual revenue and cost are precisely known.)
>
> - **Candidate Model:**
>     - **Actual** Revenue: `R_opt = 100.85` (**+0.85% gain**)
>     - **Actual** Cost: `C_opt = 15`
>     - **Actual** ROI: `ROI_opt_actual = 100.85 / 15 ≈ 6.72`
>
> The corresponding **estimated ROI range** for the candidate model is therefore: `[6.08, 7.43]`.
>
> The most extreme case is if we assume the **estimated ROI is 7.43**. According to the conclusions from Part I, we can then deduce that the **actual ROI range is [6.72, 8.21]**.
>
> Assuming the business-defined threshold **λ\* is 6**, we can confidently determine that the **Candidate Model is superior**, and deploying it would yield the **0.85% revenue gain**.
>
> **III. Conclusion:**
>
> Through the above analysis, we aim to illustrate:
> 1.  In practical application scenarios, when there is a sufficient gap between the ROI of our baseline model and the threshold λ*, the situation described above can indeed occur.
> 2.  Estimation error limits the degree to which we can approximate the optimal solution; the error bounds discussed in this paper meet fundamental industrial application standards, and our method has achieved practical results in multiple scenarios within our online system.

---

> > ### Comment · Reviewer_UjMm · 2025-11-22
> >
> > I fully understand and agree with the authors’ justification. However, I believe this further reinforces my earlier comment about reporting the actual cost differences.

---

> > > ### Author Response · Authors · 2025-11-22
> > >
> > > Thank you for the comment. As stated in our previous response, we will explicitly add the following performance metric to Section 3.3 to validate the MCET:
> > >
> > > "In our production environment, the machine cost estimated by MCET is consistently within 5% of the actual cost, a level of accuracy fully sufficient for industrial resource planning and ROI calculations."
> > >
> > > **I understand you may be suggesting to report the cost changes in these applications, and your core concern may be whether the 0.85% relative revenue increase can cover the additional investment in machine costs.**
> > >
> > > In the context of ROI-constrained model design, the application is premised upon a deliberate increase in machine cost. This is viable in scenarios where the baseline ROI provides sufficient margin above the business threshold λ\*, thereby sanctioning greater capital expenditure for incremental revenue. **The value of λ\* is set to ensure profitability, guaranteeing that the absolute revenue gain offsets the cost growth**. Owing to business confidentiality, we cannot disclose absolute costs but can offer the relative increase (e.g., ~20%). However, we maintain that the (relative) cost delta information is a scenario-specific outcome that offers limited generalizable knowledge, in contrast to the methodological paradigm we present.
> > >
> > > Additionally, for multi-scenario resource allocation, the total machine budget is fixed. The optimization involves reallocating existing resources across scenarios, not increasing overall expenditure.
> > >
> > > We hope our detailed responses have adequately addressed your concerns. Should any further questions arise, we remain available to provide additional clarification.

---

> > > > ### Comment · Reviewer_UjMm · 2025-11-22
> > > >
> > > > I sincerely thank the authors for their response and explanation. I will increase my score once all the proposed changes have been implemented in the manuscript.

---

> > > > > ### Author Response · Authors · 2025-11-23
> > > > >
> > > > > We sincerely thank you for your guidance, constructive suggestions, and your prompt feedback throughout the rebuttal process. Your input has greatly improved our manuscript. We have updated the PDF as promised and hope it meets your expectations.

---

> > > > > > ### Comment · Reviewer_UjMm · 2025-11-23
> > > > > >
> > > > > > I do not think the explanation in $R/R^*$ is clear in the updated manuscript.
> > > > > >
> > > > > > Therefore is no discussion about the cost comparison (no need to report the numbers).

---

> > > > > > > ### Author Response · Authors · 2025-11-23
> > > > > > >
> > > > > > > Dear Reviewer UjMm,
> > > > > > >
> > > > > > > Thank you for your continued engagement. We have uploaded a revised manuscript that incorporates the requested clarifications.
> > > > > > >
> > > > > > > 1) In lines 75-77, we now introduce and explain R/R* as follows:
> > > > > > >
> > > > > > > **which is evaluated on training data that is an i.i.d. sample from the full-stage online system. It directly measures, offline, the ratio between the total value (eCPM) of the top-m ads selected by the model and the total value of the ground-truth top-m ads.**
> > > > > > >
> > > > > > > We believe this provides a more intuitive and direct definition. We are fully committed to ensuring the concept is accessible and welcome any further specific suggestions you might have on the wording.
> > > > > > >
> > > > > > > 2) Your comment "Therefore is no discussion about the cost comparison (no need to report the numbers)." There may be a typo, and we think it  should be "There is no discussion about the cost comparison (no need to report the numbers)."
> > > > > > >
> > > > > > > Due to space limitations in the rebuttal version, we had to place this content in the corresponding appendix section, which you might have missed.
> > > > > > >
> > > > > > > We have added a discussion in Appendix D.3 (lines 1369-1375), emphasizing the cost changes in Case Study 1 and 2, namely, achieving further growth in online revenue by additionally investing in machine costs.
> > > > > > >
> > > > > > > We hope these revisions fully address your concerns and thank you again for your valuable feedback, which has significantly strengthened our paper.

---

> > > > > > > > ### Author Response · Authors · 2025-11-25
> > > > > > > >
> > > > > > > > Dear Reviewer UjMm,
> > > > > > > >
> > > > > > > > Thank you once again for your time and for updating your score (from 2 to 4) following our revisions. We sincerely appreciate your recognition of our efforts.
> > > > > > > >
> > > > > > > > We have noted that a score of 4, while an improvement, still represents an overall negative recommendation for acceptance.
> > > > > > > > To ensure we have addressed all your concerns, could you please let us know if any concerns remain unaddressed? We are ready to provide further clarification immediately.
> > > > > > > >
> > > > > > > > Thanks again for your guidance.

---

### Official Review · Reviewer_MBR1 · 2025-11-01

**Soundness:** 2
**Presentation:** 3
**Contribution:** 3
**Rating:** 4
**Confidence:** 3

**Summary:**

This paper investigates scaling laws in online advertisement retrieval systems, proposing an offline paradigm to predict machine costs and revenue from model configurations. The authors argue that traditional methods for finding scaling laws are prohibitively expensive in this domain, as they require numerous costly and time-consuming online A/B tests to measure the true relationship between machine cost and online revenue.

Key contributions include a offline metric R/R* that correlates with online revenue (theoretically asymptotic correlation of 1 under mild assumptions, empirically R²=0.902), fitting broken neural scaling laws (BNSL) across MLP, DSSM, and Transformer architectures in matching/pre-ranking stages, and a simulation algorithm for cost estimation.

The authors demonstrate the existence of "broken neural scaling laws" (BNSL) for various architectures (MLP, DSSM, Transformer) in their production system. They then apply this framework to two practical applications: ROI-constrained model design and multi-scenario resource allocation, claiming a substantial 5.1% combined improvement in online revenue.

**Strengths:**

High Practical Impact and Significance: The paper tackles a critical and expensive problem for any large-scale industrial ML Retrieval system: how to perform cost-aware model development and resource allocation. The ability to accurately estimate the ROI of a model configuration offline is extremely valuable. The reported +5.1% online revenue gain from applying this framework is a very strong testament to its practical utility.

Holistic and Well-Designed Framework: The authors present a complete, end-to-end paradigm. The insight to decouple the problem into a revenue-surrogate and a cost-surrogate (MCET) is clever. The MCET tool has an interesting finding that FLOPs are an unstable proxy for real-world machine cost in highly-optimized, custom serving environments.

**Weaknesses:**

Limited Conceptual Novelty: The R/R* metric is functionally a "revenue-weighted recall." Can the authors comment on the novelty of this metric in the context of prior work on utility-based or business-value-weighted metrics in recommender systems and information retrieval? The paper's strength seems to be its empirical validation rather than the novelty of the metric's formulation. The contribution is more of an engineering one—successfully applying and validating this known concept in a new domain—rather than a fundamental research one.

Theoretical guarantees of R/R* rely on "mild" assumptions (e.g., proportional pathway improvements, invariant normalized contributions) that may not hold perfectly in heterogeneous real-world ad systems, potentially limiting generalizability; more sensitivity analysis on these assumptions would strengthen claims.

Experiments are rigorous but confined to one proprietary system and standard architectures (MLP, DSSM, Transformer), lacking comparisons to diverse ad platforms or emerging recsys scaling works.

**Questions:**

For the R/R* metric, Appendix B.3 shows a sensitivity analysis for m (the cutoff) from 1 to 6, and Table 1 uses m=2. How was it chosen? Is it tuned on a validation set to maximize correlation, or does it correspond to a physical property of the system (e.g., the number of ads the Pre-ranking stage actually sends to the Ranking stage)?

The paper motivates MCET by arguing that FLOPs are a poor proxy for machine cost. However, the revenue side of the paradigm relies on FLOPs as a reliable intermediate variable (Model Config -> FLOPs -> R/R*). Why are FLOPs a stable predictor for R/R* (model performance) but not for machine cost (model speed)?

---

> ### Author Response · Authors · 2025-11-16
>
> Dear Reviewer MBR1,
>
> Thank you for your thorough and balanced review. We deeply appreciate your recognition of our work's "High Practical Impact" and "Holistic and Well-Designed Framework." We have carefully considered your valuable points and provide the following responses.
>
> **1) Regarding "Limited Conceptual Novelty" of the $R/R^\*$ metric**:
>
> We agree that the idea of value-weighted metrics exists in broader IR literature. However, we see our contribution as a novel synthesis and rigorous validation within the online advertising retrieval domain.
>
> To the best of our knowledge, this is the first work to propose such a metric and, crucially, to theoretically ground and empirically demonstrate its strong linear correlation with online business revenue in this context.
>
> The primary novelty of our paper is not any single component in isolation, but the creation of an end-to-end, actionable paradigm. We seamlessly integrate a theoretically-justified offline proxy $R/R^\*$, empirical scaling laws (BNSL), and a system-aware cost simulator (MCET) to solve the critical industrial problem of ROI-aware model development. We believe this holistic framework represents a significant step beyond applying known concepts in a new domain.
>
> **For weakness 2):**
>
> Thank you for raising this critical point. We agree that in real-world systems, these assumptions may not hold perfectly. In fact, if they did perfectly, we would expect an R² of exactly 1 in Table 1. The observed high—but not perfect—correlation (R² of 0.902 and 0.725) precisely demonstrates the metric's robustness as a superior offline proxy for online revenue, even when the theoretical assumptions are only approximately met.
>
> Furthermore, the results in Table 1 across the Pre-ranking and Matching stages can be viewed as a form of sensitivity analysis. The Matching stage, with its more complex multi-pathway architecture, represents a scenario where the "dominant pathway" assumption is more challenging. This is reflected in the lower (though still strong) R² of 0.725 compared to the Pre-ranking stage. Crucially, even in this more demanding scenario, $R/R^\*$ significantly outperforms all traditional metrics, underscoring its practical utility and generalizability.
>
> **3) Regarding "Experiments are confined to one proprietary system":**
>
> While we cannot run experiments across different companies, we designed our evaluation to maximize internal validity and generalizability: We validated our framework across three diverse model architectures (MLP, DSSM, Transformer). We tested it in two distinct business scenarios (Scene₁, Scene₂) with different user engagement patterns. We applied it to two different stages (Matching, Pre-ranking) in the cascade. This systematic variation within a large-scale production system provides strong evidence for the robustness and general applicability of our proposed paradigm.
>
> **Answers to the questions:**
>
> **Q1:** Thank you for this question. The value of m=2 was chosen based on a physical property of our system, not by tuning on a validation set. In our training data pipeline, this value corresponds to the typical number of ads which is sent to the Re-ranking stage for final selection. We will clarify this in the next version.
>
> **Q2:** The stability of FLOPs as a predictor for R/R* is an empirical finding from our scaling law experiments (Figures 3, 6, 7, 8). We observe that a model's performance is primarily governed by its total computational footprint (FLOPs), and that within a reasonably broad range, different model configurations with similar FLOPs achieve comparable R/R* scores. This empirical relationship is what makes the Model Config -> FLOPs -> R/R* link reliable for performance prediction.
>
> In contrast, machine cost (throughput/latency) is highly sensitive to specific model configurations and system-level optimizations (e.g., our first-layer optimization, kernel fusion, memory access patterns). These factors can drastically alter runtime efficiency without changing the total FLOP count, breaking a direct FLOPs-to-cost mapping. This fundamental discrepancy is precisely why we introduced the MCET—to bridge the gap between the intrinsic, architecture-agnostic computation (FLOPs) and the realized, system-specific cost.
>
> Thank you again for your constructive feedback. If any concerns remain, we are fully committed to addressing them in our follow-up responses.

---

> > ### Author Response · Authors · 2025-11-25
> >
> > Dear Reviewer MBR1,
> >
> > Thank you again for your valuable time and review.
> >
> > As we reach the midpoint of the rebuttal period, we are gently following up. We wanted to see if you had any remaining concerns after reading our response. We are, of course, ready to provide any additional clarification you might need.

---

### Official Review · Reviewer_PUhC · 2025-11-04

**Soundness:** 3
**Presentation:** 4
**Contribution:** 2
**Rating:** 6
**Confidence:** 2

**Summary:**

This paper establishes scaling trends between cost of serving an ad retrieval model vs expected online revenue for a online ad platform. More specifically, the paper makes 3 contributions - (1) it establishes R/R* (a normalized @k metric which measures the expected revenue of a model's top m predictions vs production system's expected revenue) as a robust offline proxy for online revenue as compared to standard nDCG & recall metrics; (2) it plots relationship between flops of a model vs the R/R* metric on offline data for multiple model sizes and configurations; (3) it presents practical methodology for translating a configuration of a model to its actual deployed cost which is helpful in computing ROI of a model setup. Through extensive experiments on internal closed data, the paper shows improved revenue gains when using optimal models suggested using the scaling trends established in the paper.

**Strengths:**

1. The paper is well structured and the writing is clean and easy to follow
2. The proposed methodology for establishing scaling laws is well principled
3. The paper presents proper justifications for key design choices taken

**Weaknesses:**

1. Closed evaluation - I acknowledge that the nature of the problem tackled in the paper makes it difficult to present these results in an open setting but nonetheless because of all experiments being closed source/data makes it impossible to replicate or reproduce.
2. Scope of the results - I am unsure if the results presented in the paper hold in other settings or is of interest to the ICLR community which as per my understanding focuses more on learning algorithms, architecture or a better understanding of ML models.
3. Theoretical justification assumptions are not mild - the theoretical arguments made in the paper in my opinion take a significant leap of faith (such as predicted eCPM being same as actual revenue numbers, only the top-m ads (where m~1-6) from the retrieval stages will be selected by the post-stages for exposure)

**Questions:**

1. Can the authors elaborate why MCET module is so expensive and what is a typical relation between model size and the associated machine cost predicted by the MCET?
2. Why do the authors restrict there analysis for matching and pre-ranking stage?

---

> ### Author Response · Authors · 2025-11-16
>
> Dear Reviewer PUhC,
>
> Thank you for your thoughtful review and positive feedback on our paper's structure and principled methodology. We are very encouraged by your rating. We address your specific concerns below.
>
> **For weakness 1)**, regarding "Closed evaluation" and reproducibility:
>
> We fully acknowledge the limitations of using proprietary data. To mitigate this, we have made substantial efforts in the paper and appendix to detail our methodology:
>
> ● We provided the full definition and computation process of the $R/R^\*$ metric (Section 3.1, Appendix B).
>
> ● We detailed the Machine Cost Estimation Tool (MCET) algorithm (Algorithm 1, Appendix D.2).
>
> ● We comprehensively described our experimental setups for different architectures (Appendix C).
>
> We believe this provides a clear blueprint for researchers to adapt our paradigm to other systems or domains, even if the exact replication on our data is not possible.
>
> **For weakness 2), regarding "Scope of the results" and relevance to ICLR**:
>
> We believe our work is highly relevant to the ICLR community, particularly for those interested in the practical application and scaling of ML models in real-world online advertising systems. Our core contribution is a novel framework that bridges the gap between scaling laws theory and industrial decision-making.
>
> ● We provide a practically validated paradigm for lightweight scaling law identification, which is a common challenge in deploying models under budget constraints.
>
> ● The reported 5.10% online revenue gain demonstrates a significant impact achievable through a principled, data-driven approach to model config design under budget constraints.
>
> ● We argue that this "recipe" for building actionable scaling laws under resource constraints is of direct interest to the machine learning community working on advertisement retrieval, recommender systems, and other large-scale industrial retrieval systems.
>
> **For weakness 3), regarding "Theoretical justification assumptions are not mild":**
> We thank the reviewer for this insightful comment. We agree that the term "mild" can be subjective. The ultimate validation of our assumptions, however, lies in their empirical robustness.
>
> ● Even in the presence of real-world noise and potential violations of these assumptions, the metric $R/R^\*$ demonstrated a strong and superior linear correlation (R²=0.902 and 0.725, in Table 1) with online revenue compared to all traditional metrics.
>
> ● More importantly, the predictions derived from our framework (e.g., the predicted vs. actual revenue gains of 0.33% vs. 0.38% and 0.62% vs 0.69% in Section 3.2, line 318) and the final 5.10% overall revenue improvement provide compelling evidence that our framework is robust and effective in practice.
>
> In the revised manuscript, we will address this by replacing the subjective descriptor "mild" with a focus on the objective, empirical evidence that validates our framework's effectiveness.
>
> **Answer to Questions:**
>
> **Q1:** Regarding the relationship between model size and machine cost: as discussed in Lines 363-370, while a positive correlation exists at a large scale, the specific relationship is often a complex non-linear function due to system-level optimizations. A concrete example is provided in Lines 1193-1209, which illustrates that scaling the first model layer can have a drastically different cost implication compared to scaling other layers. It is precisely this complexity that necessitates the use of our MCET for accurate, system-aware cost estimation.
>
> Regarding the cost of the MCET itself: we consider it not expensive in an industrial context. The tool requires only a single GPU and approximately 30 minutes per model configuration to produce a reliable cost estimate, without needing to train or deploy the model. This is highly efficient compared to the traditional approach, which requires full training and online deployment—a process that typically takes over 15 days to converge and stabilize in our production environment. Given that industrial settings often manage thousands or even tens of thousands of GPUs, the minimal resource overhead of the MCET makes it a cost-effective solution for rapid model iteration.
>
> **Q2**: The reasoning for restricting our analysis to these stages is detailed in the introduction (Lines 066-072) and Section 2.1 (Lines 124-132) of our paper; please refer to these sections. We will further clarify these points in the revised version to enhance readability.
>
> Thank you again for your constructive feedback. If any concerns remain, we are fully committed to addressing them in our follow-up responses.

---

> > ### Author Response · Authors · 2025-11-25
> >
> > Dear Reviewer PUhC,
> >
> > Thank you again for your valuable time and review.
> >
> > As we reach the midpoint of the rebuttal period, we are gently following up. We wanted to see if you had any remaining concerns after reading our response. We are, of course, ready to provide any additional clarification you might need.

---

### Author Response · Authors · 2025-11-23
**Updated a new revised version**

Dear reviewers,

We sincerely thank all the reviewers for their guidance and constructive suggestions, which have greatly helped us improve the manuscript. We have uploaded a revised version where the changes are highlighted in blue for your convenience.

---

### Author Response · Authors · 2025-12-02
**Final Statement to the Area Chair (Part I)**

Dear Area Chair,

We fully understand and support the emergency measures taken by the ICLR organizers following the reviewer information incident. We sincerely appreciate your role in making the final decision under these exceptional circumstances.

This statement aims to provide a clear summary of the review process for our paper. We will highlight the strengths of our work as recognized by the reviewers, detail how our rebuttal addressed their concerns, and present the constructive outcome that emerged prior to the widespread awareness of the system vulnerability.

**1. Reviewer-Recognized Strengths and Constructive Rebuttal Outcome**

First, we note that all reviewers acknowledged the core merits of our work, which formed a solid foundation for constructive discussion:
*   **High Practical Impact & Well-Designed Framework:** Reviewers highlighted the “critical and expensive problem” we tackle, the “holistic and well-designed framework,” and the “strong testament to its practical utility” represented by the **5.10% online revenue gain** (MBR1, PUhC).
*   **Important Problem & Clear Presentation:** Reviewers found the problem “important” and the initial system presentation “good” for context (ohso), and the paper “well structured” and “easy to follow” (PUhC).
*   **Interesting and Useful Work:** One reviewer explicitly stated the paper is “interesting and useful” (UjMm).

This shared recognition of the work’s value enabled a focused discussion on specific concerns, which led to a positive shift in the evaluation for two reviewers:
*   **Reviewer ohso’s** score increased from **2 (Reject)** to **8 (Accept)** on Nov 19.
*   **Reviewer UjMm’s** score increased from **2 (Reject)** to **4 (Weak Reject)** on Nov 23.

We wish to clarify the timeline: these score changes occurred **before November 27th**, which is the date the vulnerability gained widespread public attention and prompted official action. This indicates that the reviewers’ decisions to raise their scores were based on their assessment of our academic rebuttal.

**2. Summary of Rebuttal: Addressing Key Concerns**

We provided thorough, point-by-point responses to all substantive concerns, which were accepted by the reviewers:
*   **Reviewer ohso (Score: 2 → 8):** We addressed concerns about FLOPs as a scaling variable by agreeing to explicitly scope our claims to a well-defined, industrially-relevant design space (e.g., near-proportional scaling). The reviewer found this satisfactory, stating the “contribution should be accepted.”
*   **Reviewer UjMm (Score: 2 → 4):** In our detailed response, we committed to major revisions for clarity (e.g., adding intuitive explanations, formalizing theorems). Regarding the critical question on cost comparison, we provided a **rigorous mathematical analysis** and a **concrete example**, which led the reviewer to explicitly state, “I fully understand and agree with the authors’ justification” and subsequently raise the score. Although the rebuttal process was terminated before the reviewer could confirm the final revisions to the introduction, the core methodological concerns were resolved to the reviewer's satisfaction. The reviewer's primary remaining concern seems to be the intuitive explanation of the $R/R^\*$ metric in the introduction.
*   **Reviewer MBR1 (Score: 4):** The reviewer recognized the “High Practical Impact” of our work but expressed three core concerns: (1) limited conceptual novelty of the $R/R^\*$ metric, (2) generalizability of theoretical assumptions, and (3) experimental scope confined to one proprietary system. In our rebuttal, we systematically addressed each point: (1) We clarified that the conceptual novelty **extends beyond any single component**. It includes **the first definition and thorough validation (both theoretical and empirical) of the $R/R^\*$ metric using full-stage training samples** as a robust offline proxy for online revenue. Crucially, the primary contribution is **the creation of the first end-to-end paradigm** that integrates this theoretically-grounded metric, empirical scaling laws, and a system-aware cost simulator to solve a critical industrial problem. (2) We argued that the high empirical $R^2$ values (0.902 and 0.725) themselves demonstrate the framework’s robustness even when theoretical assumptions are not perfectly met, which is a common scenario in real-world systems. (3) We highlighted the internal diversity of our validation (three architectures, two business scenarios, two system stages) as strong evidence for the paradigm’s general applicability within complex industrial environments.
*   **Reviewer PUhC (Score: 6):** We emphasized that our detailed methodology serves as a **reproducible blueprint**, and that our work provides a validated **“recipe”** for both **the identification of actionable scaling laws in ad retrieval systems** and, crucially, **the subsequent ROI-aware optimization of model configurations under budget or ROI constraints**.

---

> ### Author Response · Authors · 2025-12-02
> **Final Statement to the Area Chair (Part II)**
>
> **3. Core Value of Our Work**
>
> Through this process, the value of our paper is clear:
> 1.  **High Practical Impact:** Achieves a **verified 5.10% online revenue increase**.
> 2.  **Methodological Contribution:** Proposes the **first lightweight, offline paradigm** bridging scaling laws with online revenue and budget for advertisement retrieval.
> 3.  **Rigorous Revision Process:** The rebuttal process facilitated a **thorough academic exchange**. We integrated all feasible feedback, resulting in a **substantially improved manuscript** with enhanced clarity, formalized theorems, and a more focused discussion of contributions and limitations.
>
> We respectfully request that your final decision be based on the **academic quality** of our paper, its **demonstrated impact**, and the successful resolution of concerns during the discussion. Thanks again!
>
> Sincerely,
>
> The Authors of Paper #16539

---

### Meta-Review · Area_Chair_HhdE · 2026-01-04

**Summary:**

## Summary
This paper proposes a lightweight paradigm for identifying scaling laws in online advertisement retrieval systems. The core contributions include: (1) a novel offline metric R/R* that correlates with online revenue, (2) empirical validation of broken neural scaling laws (BNSL) across MLP, DSSM, and Transformer architectures, and (3) a Machine Cost Estimation Tool (MCET) for system-aware cost prediction. The authors demonstrate practical applications achieving a reported 5.10% online revenue gain.

## Overall Score

ohso: 2 → 8 (tend to accept after rebuttal)

UjMm: 2 → 4 (improved but still negative; main remaining friction on clarity/cost-comparison discussion)

MBR1: 4 (no response during the discussion)

PUhC: 6


## Addressed Concerns
* Most concerns regarding clarity have been resolved, including details of the experiments, the definition of hard permutation matrix, etc. (MBR1, UjMm, ohso)
* Typos and errors (UjMm)
* Further explanation on the nature of scaling laws (UjMm, ohso)

## Remaining Controversial Concerns

* Concerns about FLOPs as a scaling variable (ohso)
* The clarity of the key concept R/R* (UjMm)
* Scope and reproducibility in other domains. (MBR1)
* Supplementary theoretical assumptions for R/R* (MBR1, UjMm, PUhC)

## Conclusion:
Overall, during the discussion, two reviewers (ohso and UjMm) were partly satisfied with the authors' responses, as reflected in their score changes (2-8 and 2-4, respectively). Reviewer UjMm still has some reservations about the writing clarity and certain definitions (R/R*). Reviewer MBR1 also voiced concerns about the paper's novelty and generalizability.
This paper has potential to be stronger with clearer writing and a deeper discussion on scaling laws in the context of Ads.

**Reviewer Concerns:**

Refer to Summary

**Reviewer Scores:**

Refer to Summary

---

### Decision · Program_Chairs · 2026-01-26

Reject